# A transient protein folding response targets aggregation in the early phase of TDP-43-mediated neurodegeneration

Rebecca San Gil [1], Dana Pascovici [2,3], Juliana Venturato [1], Heledd Brown-Wright [1], Prachi Mehta[1,4], Lidia Madrid San Martin[1], Jemma Wu[3], Wei Luan[1], Yi Kit Chui[1], Adekunle T. Bademosi [1], Shilpa Swaminathan[1], Serey Naidoo[5,6], Britt A. Berning[1], Amanda L. Wright[1], Sean S. Keating[1], Maurice A. Curtis [6,7], Richard L. M. Faull[6,7], John D. Lee [8], Shyuan T. Ngo [9], Albert Lee [4], Marco Morsch [4], Roger S. Chung[4], Emma Scotter [5,6], Leszek Lisowski [10,11,12], Mehdi Mirzaei[3] & Adam K. Walker [1] ✉

Understanding the mechanisms that drive TDP-43 pathology is integral to combating amyotrophic lateral sclerosis (ALS), frontotemporal lobar degeneration (FTLD) and other neurodegenerative diseases. Here we generated a longitudinal quantitative proteomic map of the cortex from the cytoplasmic TDP-43 rNLS8 mouse model of ALS and FTLD, and developed a complementary open-access webtool, TDP-map (https://shiny.rcc.uq.edu.au/TDP-map/). We identified distinct protein subsets enriched for diverse biological pathways with temporal alterations in protein abundance, including increases in protein folding factors prior to disease onset. This included increased levels of DnaJ homolog subfamily B member 5, DNAJB5, which also co-localized with TDP-43 pathology in diseased human motor cortex. DNAJB5 over-expression decreased TDP-43 aggregation in cell and cortical neuron cultures, and knockout of Dnajb5 exacerbated motor impairments caused by AAV-mediated cytoplasmic TDP-43 expression in mice. Together, these findings reveal molecular mechanisms at distinct stages of ALS and FTLD progression and suggest that protein folding factors could be protective in neurodegenerative diseases.

Dysfunction of RNA-binding proteins (RBPs) is implicated in several neurodegenerative diseases, including amyotrophic lateral sclerosis (ALS) and frontotemporal lobar degeneration (FTLD)[1]. RBPs contain intrinsically disordered regions and are particularly prone to protein misfolding and aggregation[2–5], leading to their deposition into inclusions in the brain and/or spinal cord in ALS and FTLD. Mutations in RBP-coding genes, such as *TARDBP* that encodes TAR DNA-binding protein 43 (TDP-43), can cause ALS and FTLD in a small minority of cases[1]. However, in 95% of ALS and 50% of FTLD cases, non-mutated wild-type TDP-43 is depleted from the nucleus and accumulates in the cytoplasm where it becomes misfolded, post-translationally modified,

and forms cytoplasmic inclusions[6,7]. This process strongly correlates with neuronal degeneration[8–11].

Initial evidence describes the complex and dynamic nature of the biology of TDP-43 dysfunction in vivo. For example, layer V motor cortex neurons display increased hyperexcitability and altered abundance of select glutamate receptors when TDP-43 is expressed in the cytoplasm[12,13]. Proteostasis components, heat shock proteins (HSPs) HSPB1[14,15], HSP70 and HSP40[16,17], are significantly decreased and/or sequestered into inclusions in post-mortem sporadic ALS tissue. In addition, mediators of the integrated stress response are up-regulated in the cortex of mice over-expressing cytoplasmic TDP-43

(rNLS8 mice[18]) prior to disease onset and in early disease[19]. However, the chronology of mechanisms that drive and respond to progressive TDP-43 dysfunction in early disease stages remains largely unclear. Integral to the continued advancement of this field is a better understanding of the mechanisms that drive disease and endogenous protective mechanisms that are activated in response to cytoplasmic TDP-43 such that they can be harnessed in the design of therapeutic strategies.

Here, we sought to develop a global unbiased map of the biochemical signatures driving onset, early, and late stages of TDP-43 proteinopathies. Cortex tissues of rNLS8 mice[18] were analyzed using quantitative proteomics at multiple timepoints prior to disease onset and throughout disease progression, as well as during the recovery phase following the doxycycline (dox)-dependent inhibition of cytoplasmic TDP-43 expression. Weighted correlation network analysis (WCNA) revealed eight subsets of longitudinally correlated proteins, of which five subsets corresponded to distinct patterns of time-dependent and disease-relevant changes that involved alterations to numerous biological processes. A subset of protein folding factors were specifically increased in neurons of rNLS8 mice prior to and at the earliest stages of disease but not at later timepoints, suggesting potential disease-associated suppression of early protective responses. One such protein folding factor, the HSP40 family member DNAJB5, decreased pathological TDP-43 aggregation in cells and neuron culture models, and co-localized with TDP-43 inclusions in human motor cortex. Conversely, knockout of *Dnajb5* exacerbated motor

impairments in mice with AAV-mediated cytoplasmic TDP-43 pathology. Late-stage rNLS8 mouse protein signatures strongly correlated with those of human post-mortem ALS and FTLD-TDP tissues, indicating validity of this model for understanding mechanisms of human disease progression. Therefore, our work reveals multiple biological pathways that display temporal patterns of activation and repression throughout neurodegenerative disease, including a cluster of protein folding factors that may be elevated to combat protein aggregation in the earliest stages of TDP-43 proteinopathies.

## Results

### Dynamic molecular programs define disease stages in rNLS8 mice

ALS and FTLD are progressive diseases with protracted courses in which molecular initiation of neurodegeneration occurs prior to disease onset[20]. To identify global protein changes driving disease, we profiled changes in protein abundance in the rNLS8 cortex at pre-onset (1 wk), onset (2 wk), early (4 wk) and late (6 wk) disease and dox-dependent recovery (+rec; Fig. 1a). The rNLS8 mice at pre-onset (1 wk) show dox-induced neuron-specific expression of the human cytoplasmic TDP-43 transgene and loss of endogenous mouse TDP-43, at onset (2 wk) motor deficits emerge, at early disease (4 wk) there is a decrease in muscle innervation, cortical neuron degeneration and astrogliosis, at late disease (6 wk) there is microglial dysfunction[21], spinal cord motor neuron degeneration and a dramatic motor deficit[18]. Mice in recovery (6 wk off dox and 2 wk on dox; +rec) show

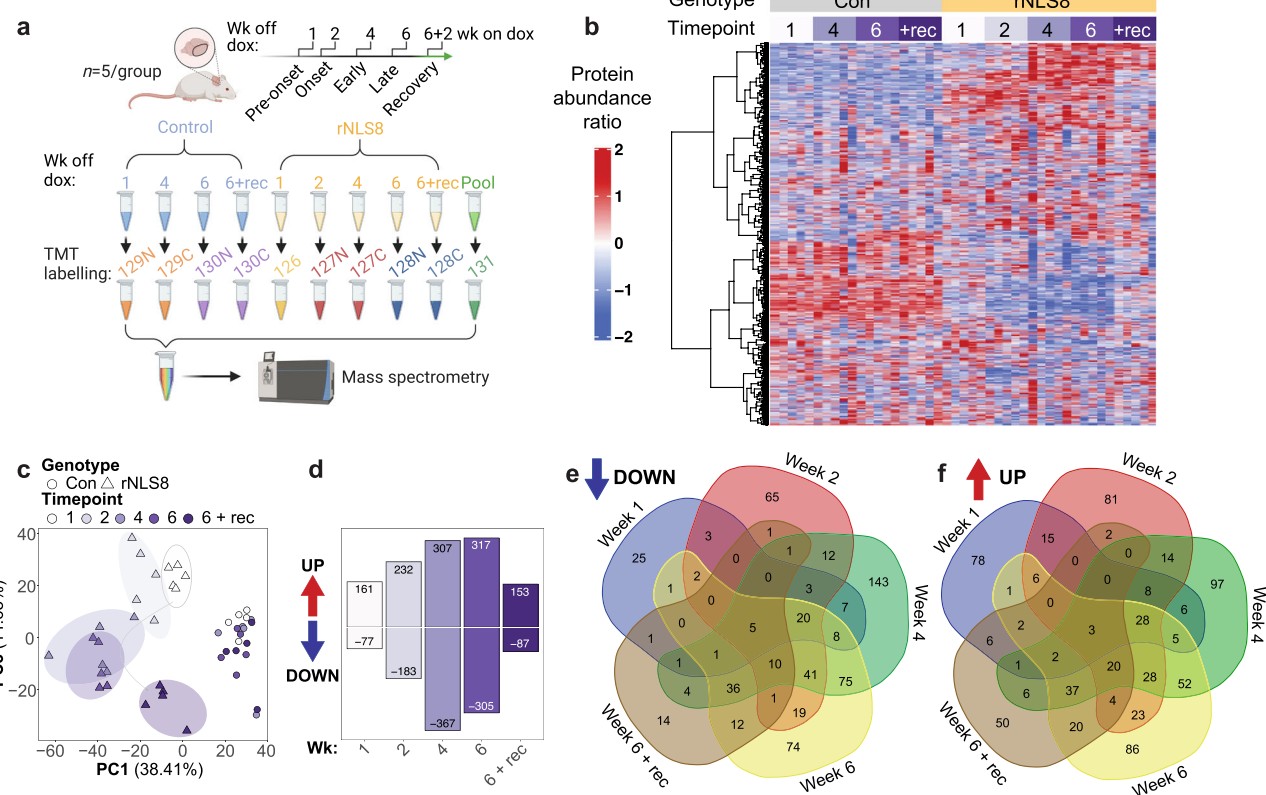

**Fig. 1 | Longitudinal proteomics of rNLS8 mice reveals global changes in protein profiles in the cortex at pre-onset, onset, early, and late disease, and in recovery.** a Schematic of experiment design and tandem mass tag (TMT) labeling. **b** Heat map of the protein abundance ratio of quantified proteins in control (Con) and rNLS8 mice at 1, 2, 4, and 6 weeks off doxycycline (dox) and mice in recovery (+rec; 6 wk off dox and 2 wk on dox). Each column represents data from an individual mouse (*n* = 5 per group). **c** Principal component (PC) analysis dimensions PC1 and PC3 (proportion of variation from top 5 PC in parentheses). **d** Number of

proteins significantly increased or decreased in rNLS8 mice at each timepoint compared to controls. Significantly altered proteins were identified using two-sided student *t* tests of log-transformed ratios (*P* < 0.05, fold change >1.2; Supplementary Data 2). **e** Venn diagram showing the number of significantly decreased proteins that are common and unique to each time point in rNLS8 mice compared to controls. **f** Venn diagram showing the number of significantly increased proteins that are common and unique to each time point in rNLS8 mice compared to controls. Source data are provided as a Source Data file.

clearance of cytoplasmic TDP-43, restoration of endogenous TDP-43 in neurons, muscle re-innervation from remaining neurons, and regain motor skills[18].

High-coverage labeling proteomics allowed quantification of $n = 6569$ proteins across samples and revealed clusters of proteins with disease-specific longitudinal changes in abundance relative to control mice (Fig. 1b, Supplementary Data 1). Principal component analysis (PCA) demonstrated clear clustering of control samples (PC1 defined by mouse genotype), and distinct grouping of rNLS8 mouse cortex samples at each disease stage (PC3 defined by time-point), revealing the progression of molecular signatures (Fig. 1c, Supplementary Data 1, Supplementary Fig. 1). The rNLS8 mouse cortex in recovery also exhibited a PCA sample cluster distinct from the control and pre-onset rNLS8 samples, suggesting that the cortex of rNLS8 mice during recovery displays a combination of lasting and reversible biochemical changes (Fig. 1c). Analysis of proteins that were significantly increased or decreased in abundance at each timepoint revealed accelerating changes, with the highest numbers of alterations detected at later timepoints, followed by partial normalization in recovery mice (Fig. 1d, Supplementary Data 2). A total of $n = 463$ proteins ($n = 229$ increased and $n = 240$ decreased in abundance) were restored to control levels in the cortex in recovering animals compared to late disease, indicating rapid resolution of some molecular signatures of disease. Comparisons of the proteins that were altered in abundance at each timepoint further identified core groups of common protein signatures corresponding to changes in early and late disease, and recovery (Fig. 1e, f, Supplementary Data 3). Additionally, distinct sets of protein alterations were detected at single timepoints of disease (Fig. 1e, f). Overall, top-level analysis of these datasets indicated disease-stage-related molecular signatures associated with progressive neurodegeneration in the cortex of rNLS8 mice over time.

To further investigate the molecular alterations that characterize each stage of disease, we next focused on the proteins that were significantly increased or decreased in abundance in the rNLS8 cortex at each time point. As expected[18], TDP-43 levels were increased after 1, 2, 4, and 6 weeks off dox and returned to control levels in recovery (Fig. 2a–e, volcano plots). In rNLS8 mice prior to disease onset (1 wk), $n = 77$ proteins were decreased and $n = 161$ proteins were increased compared to control mice, indicating very early biochemical changes even before the emergence of motor deficits[18]. Proteins that were decreased in abundance prior to disease onset were enriched for those that mediate chemical transmission at the synapse, synapse organization, and axonogenesis (including serine/threonine protein kinase DCLK1 and stathmin STMN1), indicating very early neuronal deficits prior to overt disease phenotypes (Fig. 2a left). A significant proportion of increased proteins in pre-onset animals were enriched for components of actin filament-based movement, regulation of macroautophagy, and mitochondrial respiratory chain complex 1 (mitochondrial import receptor subunit TOM34 and voltage-dependent anion-selective channel protein 3 VDAC3; Fig. 2a right, and 2f protein-protein interaction network).

Analysis of a subset of proteins that have previously been associated with ALS (Fig. 2a–e; highlighted in gold, TDP-43[6], ataxin 2[22–25], stathmin 2[26–29], UNC13A[30–33], and TTBK1[34–36]) were tracked to monitor for changes in abundance throughout the course of disease. For example, levels of stathmin-2 (STMN2) were significantly decreased in rNLS8 mice at 1, 2, 4, and 6 weeks off Dox, with normalization in recovery mice (>1.2 FC decrease for STMN2 at 2 weeks). In addition, levels of UNC13A were significantly decreased in rNLS8 mice at 2, 4, 6 weeks and in recovery mice (maximum of 0.87-fold change at 6 weeks off Dox). Regarding UNC13A, decline in protein abundance may be the result of cryptic exon inclusion in *Unc13a* transcripts, although TDP-43 binding occurs at different sites in mouse compared to the human transcript[37]. Changes to STMN2 and UNC13A in the rNLS8 cortex may also reflect alterations to protein rather than transcript, since TDP-43 regulation of these sites are not conserved from human

to mouse[29,30]. Indeed, the decline in STMN2 and UNC13A occurs alongside a decrease in abundance of other proteins associated with axonogenesis and chemical synaptic transmission, respectively.

At disease onset (2 wk) $n = 183$ proteins were decreased and $n = 232$ proteins were increased in abundance in rNLS8 compared to control cortex, showing advancing proteome disruption. The decreased proteins at 2 weeks were enriched for proteins associated with membrane depolarization during action potentials, chemical synaptic transmission, and neuron projection morphogenesis (including inositol 1,4,5-triphosphate receptor type 1 ITPR1 and 14-3-3 protein eta YWHAH; Fig. 2b, left). The increased proteins at disease onset were significantly enriched for lysosomal proteins, protein processing in the endoplasmic reticulum, regulation of intracellular protein transport, mRNA processing, and protein folding (including DnaJ homolog subfamily B member 5 DNAJB5 and peptidyl-prolyl cis-trans isomerase FKBP4; Fig. 2b, right), suggesting effects on protein homeostasis in early disease. One protein of interest was wolframin-1 (WFS1), which showed significantly decreased levels in rNLS8 mice from 2 weeks off dox. WFS1-positive neurons are specifically affected in a FTLD-tau mouse model[38], suggesting that loss of WFS1 protein in the rNLS8 cortex may reflect specific neuronal vulnerabilities related to FTLD-linked neurodegeneration.

At an early disease stage (4 wk), $n = 367$ proteins were decreased and $n = 307$ proteins were increased in abundance in rNLS8 mice, revealing even more widespread disease-relevant changes than at previous timepoints. The decreased proteins were significantly enriched for proteins strongly implicated in neurodegeneration including the KEGG pathways of Alzheimer's disease, Huntington's disease, and ALS. The decreased proteins were also enriched for neuronal system proteins (including calcium/calmodulin-dependent protein kinase CAMKK2 and glutamate receptor 3 GRIA3), which are largely involved in chemical synaptic transmission, highlighting the predominance of neurodegeneration from this timepoint in the rNLS8 mice (Fig. 2c left). Increased proteins were enriched in those involved in neutrophil degranulation (likely driven by alterations in proteins involved in lysosomal function, which was listed as a secondary term), complement activation (including complement C4b and cathepsin S CTSS), and protein folding (Fig. 2c right), suggesting activation of a diverse set of biological processes involving both adaptive and degenerative pathways. RNAseq of microglia isolated from the rNLS8 mouse cortex after 4 weeks off dox[39] has previously revealed a significant increase in genes enriched for neutrophil degranulation, suggesting that the detection of this feature in our proteomics analysis may be indicative of microglial activation.

In late disease (6 wk), $n = 305$ proteins were decreased and $n = 317$ proteins were increased in abundance in the rNLS8 cortex compared to controls, demonstrating continued widespread biochemical changes at this late-disease stage that is characterized by advanced motor deficits[18]. Decreased proteins were involved in pathways that likely reflect neurodegenerative loss of neurons, including chemical synaptic transmission (including ephrin type-A receptor 4 EPHA4 and synaptosomal-associated protein 25 SNAP25; Fig. 2d left and g), synapse organization, ion transport, and mitochondrial respiratory chain complex I. Notably, a subset of the respiratory chain complex I proteins that was increased in pre-onset rNLS8 mice was decreased in late disease rNLS8 mice, suggesting differential alterations in mitochondrial dysfunction across the disease course. This decrease at late disease stages is in line with findings from analysis of the sciatic nerve axoplasm of the hTDP-43$^{\Delta NLS}$ mouse model, which demonstrated dramatic changes to mitochondrial function in neurons with cytoplasmic TDP-43[40]. Increased proteins in late disease were enriched in neutrophil degranulation (Fig. 2d right and h) and ubiquinone biosynthesis. Regulation of protein import into the nucleus was also a notable pathway with increased protein abundance at this timepoint (including importin subunit alpha-3 KPNA4 and prelamin-A/C LMNA; Fig. 2d),

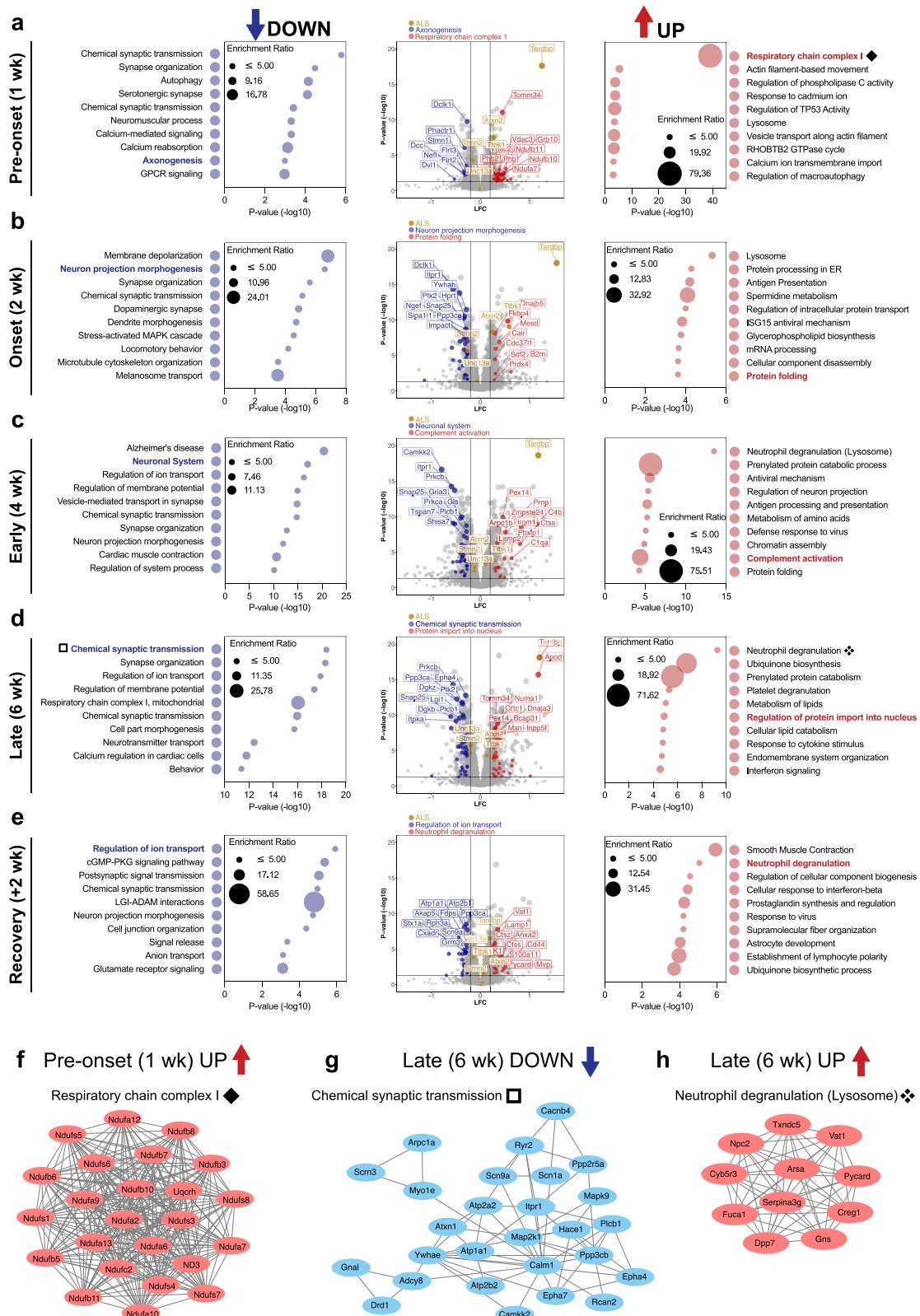

suggesting that accumulation of cytoplasmic TDP-43, even with its ablated functional NLS, may directly impact nuclear import.

Inhibition of hTDP-43$^{\Delta NLS}$ expression for 2 weeks enables the rNLS8 mice to functionally recover after reaching a late disease stage (6 wk off dox + 2 wk recovery on dox)[18]. Of the $n = 87$ proteins that were decreased in abundance in recovery compared to grouped control animals, $n = 65$ were also decreased in late disease. Of the $n = 153$ proteins increased in abundance in recovery, $n = 88$ were also increased in late disease. This indicates the likely continuation of many disease-associated alterations in recovery animals. Mice in recovery also shared only $n = 8$ decreased and $n = 14$ increased proteins in common with pre-onset mice (1 wk), indicating that distinct biochemical processes mediate recovery and disease initiation. In recovery, only $n = 50$ proteins were increased in the rNLS8 mouse cortex.

**Fig. 2 | Gene ontology analysis reveals alterations of key biological processes driven by TDP-43-mediated disease. a**–**e** Metascape gene ontology of significantly decreased (*left*) and increased proteins (*right*) with the size of the bubble being dependent on the enrichment ratio of each term. Volcano plots (*middle*) of mean log fold change (LFC; rNLS8/Con) and significance level (*P* value (−log10)). Proteins belonging to selected disease-relevant gene ontology terms at each timepoint have been highlighted. The protein subset that is decreased is shown in blue and increased in red. A select subset of ALS-associated proteins has been highlighted in gold. The top 10 proteins, ranked by *P* value, in each highlighted gene ontology subset of proteins have been labeled. Significantly altered proteins were identified based on two-sided student *t* tests of log-transformed ratios ($P < 0.05$, fold change >1.2). **a** Pre-onset (1 wk), (**b**) onset (2 wk), (**c**) early disease (4 wk), (**d**) late disease (6 wk) and (**e**) recovery (6 wk + 2 wk on dox recovery) mice. **f** Protein–protein interactions of components of the mitochondrial respiratory chain complex 1, which are increased in pre-onset rNLS8 mice (1 wk, **a**). **g** Protein–protein interaction network of components of chemical synaptic transmission, which are significantly decreased in late-disease rNLS8 mice (6 wk, **d**). **h** Protein–protein interactions of components of neutrophil degranulation (lysosomes), which are increased in late-disease mice (6 wk, **d**). Source data are provided as a Source Data file.

These were enriched in proteins involved in ensheathment of neurons, including the proteins myelin oligodendrocyte protein (MOG), tetraspanin-2 (TSPAN2), claudin-11 (CLDN11), junctional adhesion molecule C (JAM3), and plasmolipin (PLLP), suggesting that changes in oligodendrocyte populations and myelination may be particularly important in disease recovery. Despite functional motor recovery at this timepoint[18], proteins associated with a subset of neuronal pathways including chemical synaptic transmission and cell projection morphogenesis remained decreased (Fig. 2e, left), similar to late-disease mice. All $n = 153$ increased proteins in recovery were significantly associated with pathways such as smooth muscle contraction, the cellular response to interferon-beta, and astrocytic development (Fig. 2e right), potentially representing adaptive changes across diverse biological pathways. Overall, these findings suggest that there is irreversible loss of neurons in the rNLS8 mouse cortex in late disease, which continues to be reflected at the recovery timepoint. However, functional recovery involves the reversal of a proportion of late-disease changes, including some neuronally-expressed proteins, and is also associated with a unique program of myelination proteins and glial cell activation.

## Distinct protein modules show differential disease-specific abundance profiles over time in the rNLS8 mouse cortex

To gain a detailed understanding of the molecular mechanisms that initiate and drive disease, we employed WCNA[41] of the mouse quantitative longitudinal proteomics dataset. This strategy clusters proteins into subsets, herein referred to as modules, based on their time-dependent patterns of correlated protein abundance in control and rNLS8 samples. Of the $n = 6569$ proteins quantified, $n = 977$ were assigned to a module, and eight distinct modules were defined. The eigenprotein network dendrogram and adjacency heat map demonstrated the relatedness of the modules to one another (Fig. 3a and Supplementary Fig. 2a). Three of the modules showed age-associated changes (pink, red, and green modules), including changes in both control and rNLS8 mice across time, suggesting less relevance of these modules to TDP-43 pathology (Supplementary Fig. 2b–d). However, this approach also revealed five disease-related protein modules that clearly showed changes in protein abundance exclusively in rNLS8 mouse cortex with different disease stages and are enriched for distinct biological processes (Table 1 and Fig. 3b–f).

Each disease-relevant module was enriched for different biological processes showing that different categories of proteins have distinct temporal expression profiles (Table 1). For example, black module proteins are, at least in part, involved in ensheathment of neurons, enriched for oligodendrocytes and are increased in the cortex in late disease and recovery (Table 1, Supplementary Fig. 3). Whereas, turquoise module proteins are involved in neuronal systems such as chemical synaptic transmission, are decreased in abundance starting at disease onset, and partially reversed in recovery (Supplementary Fig. 4).

## Neurons, oligodendrocytes, and microglia are enriched in disease-specific modules

To deconvolve the whole cortex proteomics data and identify the cell-autonomous and non-cell-autonomous contributions to the proteomics signatures observed, we conducted cell-type enrichment analysis of the WCNA modules. The protein lists from each module were compared to cell-type-specific protein sets[42]. Notably, the black module was enriched for oligodendrocyte proteins, reflecting the alterations in the ensheathment of neurons in the rNLS8 cortex in recovery (Fig. 3g, h and Supplementary Fig. 3). In contrast, the brown module correlated with microglial proteins, suggesting neuroinflammatory changes in late disease (Fig. 3g, i and Supplementary Fig. 5). Proteins in the turquoise module exhibited a time-dependent decrease and normalization towards control levels in recovery and were enriched in neuronal proteins (Fig. 3g, j). This suggests that the changes in this module reflect cell-intrinsic proteins related to the accumulation of TDP-43 pathology, with decreased protein abundance not solely due to neuron loss. The turquoise module contained $n = 38$ neuronal proteins, including calcium/calmodulin-dependent protein kinase type IV (CAMK4), sphingomyelin phosphodiesterase 3 (SMPD3), and GTP-binding protein Di-Ras2 (DIRAS2), three of the most significantly depleted neuronal proteins in late-disease rNLS8 cortex (Fig. 3j). The turquoise module was defined by a decrease in proteins involved in the neuronal system Reactome pathway, and several gene ontology terms for chemical synaptic transmission, and the ALS KEGG pathway (Supplementary Fig. 4). The blue (Supplementary Fig. 6) and magenta modules showed no enrichment of cell-specific proteins. Together, cell-type-enrichment analysis of the WCNA modules suggests both cell-autonomous involvement of neurons and non-cell-autonomous involvement of oligodendrocytes and microglia in TDP-43-mediated disease and recovery.

## The protein folding module (magenta) defines an early transient protective response in rNLS8 mice

Amongst the eight defined modules, proteins in the magenta module exhibited a unique pattern of change, being increased in pre-disease and onset rNLS8 mice but unaltered in early disease and late disease (Figs. 3f, 4a). We observed no change in magenta module proteins in control mice across time or in rNLS8 mice during recovery compared to controls (Figs. 3f, 4a). This is suggestive of a specific response involved in early disease processes that was not maintained throughout later disease stages.

Notably, the proteins in the magenta module were defined by their involvement in the HSP90 chaperone cycle, protein folding, and regulation of protein binding, suggesting a highly relevant subset of proteins for ALS and FTLD, which are protein misfolding disorders (Fig. 4b). Seven of the proteins are members of a protein–protein interaction network, suggesting a coordinated function in disease, namely, HSP 90 alpha family class B member 1 (HSP90AB1), HSP90 co-chaperone cell division cycle 37 (CDC37), DnaJ homolog subfamily B member 5 (DNAJB5), calreticulin (CALR), stress-induced phosphoprotein 1 (STIP1), translocase of outer mitochondrial membrane 34 (TOMM34), and cytoplasmic dynein 1 intermediate chain 2 (DYNC1i2) (Fig. 4c). Except for DYNC1i2 and TOMM34, these proteins are either chaperones or co-chaperones that act in concert to bind misfolded protein substrates for delivery to HSP90 or HSP70 ATP-dependent protein re-folding cycles[43–45].

We next validated the changes in protein abundance of magenta module proteins as identified by the proteomic analysis.

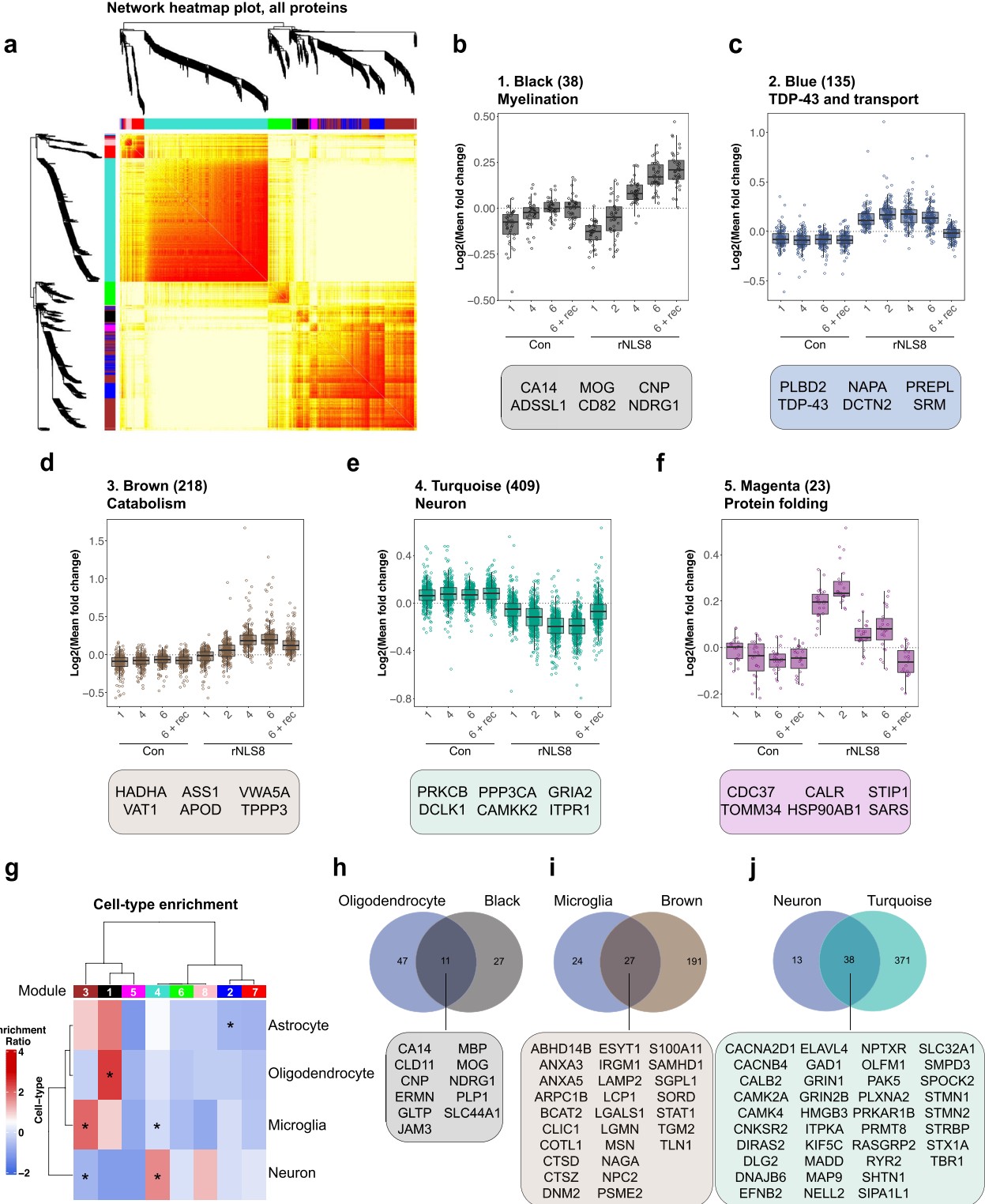

Immunoblotting confirmed that there was a significant increase in HSP90AB1, DNAJB5, and CDC37 protein levels in pre-onset and disease onset mice, with the protein levels of these three proteins returning to control levels in early disease (Fig. 4d–f, Supplementary Fig. 7). The DNAJB5 antibody showed multiple bands by immunoblot, with the indicated 3 bands being specific to DNAJB5 (Supplementary Fig. 8). There was no change in the levels of these proteins in the separated insoluble protein fraction of rNLS8 cortex, suggesting that the

decrease back to control levels from 4 weeks off dox was due to decreased total protein levels rather than sequestration from the soluble to the insoluble fraction (Supplementary Fig. 9). The mRNA levels of *Hsp90ab1*, *Dnajb5*, *Cdc37* and *Calr* were not significantly increased in the rNLS8 cortex, and therefore did not correlate with the increased protein levels in pre-onset mice (Fig. 4g and Supplementary Fig. 10). This indicates that a post-transcriptional mechanism regulates the changes in the levels of these proteins at different disease stages.

**Fig. 3 | Weighted correlation network analysis of longitudinal quantitative proteomics of rNLS8 mouse cortex reveals specific modules of proteins that correlate with distinct disease stages. a** Hierarchical clustering based on topology overlap measurement distance reveals 8 modules by WCNA. **b**–**f** Log fold change of protein abundance in control (Con) and rNLS8 cortex at each timepoint for each subset of module proteins. **b** Black, (**c**) blue, (**d**) brown, (**e**) turquoise, and (**f**) magenta modules are disease-relevant and show changes in protein abundance in the rNLS8 cortex only. The number of proteins in each module is listed in parentheses together with the top gene ontology term. Featured proteins from each module represent the top 6 proteins ranked by module membership (kME) value. All data are from $n = 5$ mice per group and each data point (open circles) represents the mean log fold change of each protein in each module. Box and whisker plots in (**b**–**f**) show a line at the median, upper and lower hinges are the first and third quartiles, and the top and bottom whiskers extend to the largest and lowest value no more than 1.5 * the inter-quartile range, respectively. **g** Cell-type enrichment analysis of modules with astrocytic, microglial, oligodendrocyte, and neuronal protein lists derived from published datasets[42]. Modules were identified as significantly enriched or depleted of cell-specific proteins using a Fisher's exact test. **h** Overlap of all oligodendrocyte-specific and black module proteins. **i** Overlap of all microglial-specific and brown module proteins. **j** Overlap of all neuronal-specific and turquoise module proteins. All proteins listed are in alphabetical order. Source data are provided as a Source Data file.

## Table 1 | Summary of disease relevant modules

| Module | # Proteins | Time-resolved change | Cell enrichment | Top GO term | Top 3 proteins |
|---|---|---|---|---|---|
| 1. Black | 38 | Late disease increase, increased in recovery | Oligodendrocyte ($P = 5.2e-6$) | Ensheathment of neurons ($P = 8.1e-7$) | CA14 ADSSL1 MOG |
| 2. Blue | 135 | Early maintained disease increase, control levels in recovery | - | ER to Golgi anterograde transport ($P = 1.5e-8$) | PLBD2 TDP-43 NAPA |
| 3. Brown | 218 | Late disease increase, partially reversed in recovery | Microglia ($P = 3.5e-5$) | Ubiquinone biosynthetic process ($P = 1.1e-8$) | HADHA VAT1 ASS1 |
| 4. Turquoise | 409 | Decreased in disease, partially reversed in recovery | Neuron ($P = 1.5e-6$) | Neuronal system ($P = 4.3e-29$) | PRKCB PCLK1 PPP3CA |
| 5. Magenta | 23 | Increased at pre-onset and onset, partially reversed in early and late, control levels in recovery | - | HSP90 chaperone cycle ($P = 2.0e-5$) | CDC37 TOMM34 CALR |

The number of proteins (# proteins) that are members of each module, the protein abundance profile (time-resolved change) across time in control and disease mice, the main contributing cell type (cell enrichment), the top gene ontology (GO) term shows the biological processes that define each module, and the top 3 proteins with highest module membership values (kME). Two-sided Fisher's exact test to determine whether cell-type enrichment was statistically significant within each module ($P < 0.05$). Protein sets were analyzed using Metascape[108] where significantly enriched gene ontology terms of each module were identified using the hypergeometric test and Benjamini-Hochberg $P$ value correction algorithm.

Increased protein levels of DNAJB5, HSP90AB1, CDC37 and CALR were observed in neurons of the primary motor cortex with cytoplasmic TDP-43 accumulation (Fig. 4h).

To determine whether HSP90AB1, DNAJB5, and CDC37 protein levels were increased in other cell and animal models of ALS, we analyzed HEK293 cells expressing cytoplasmic TDP-43 variants and the cortex and spinal cord of TDP-43$^{Q331K}$ mice. In HEK293 cells there was a trend towards increased levels of DNAJB5 with over-expression of all TDP-43 variants, TDP-43$^{ΔNLS}$ decreased levels of HSP90AB1, and levels of CDC37 were unaffected in cells with cytoplasmic TDP-43 but increased with TDP-43$^{WT}$ (Supplementary Fig. 11). We found that the over-expression of different cytoplasmic TDP-43 variants had a variable effect on the abundance of each of these chaperones in HEK293 cells after 48 h (Supplementary Fig. 11). Regarding the TDP-43$^{Q331K}$ transgenic mouse, there was no change in the abundance of DNAJB5, HSP90AB1, or CDC37 in the cortex or spinal cord compared to littermate controls at 3 months, a timepoint consistent with pre-onset disease stage[46,47], nor in the cortex at 16 months, a mid-disease stage (Supplementary Fig. 12). The differences in protein folding responses between the rNLS8 and TDP-43$^{Q331K}$ mice is not surprising, given that they may model different aspects of the pathobiology of ALS. The TDP-43$^{Q331K}$ mouse shows gain and loss of normal TDP-43 function (aberrant splicing) but not cytoplasmic TDP-43 inclusions and develops late-onset motor deficits that do not progress to a typical ALS-like disease end-stage[46,48]. In contrast, the rNLS8 mouse shows a decrease in endogenous TDP-43, suggesting potential loss of normal nuclear function, in addition to cytoplasmic TDP-43 inclusions, neurodegeneration, neuroinflammation, and shortened lifespan[18], however, whether aberrant splicing of TDP-43 target transcripts occurs in rNLS8 mice remains to be established.

To determine whether rNLS8 mice show a global alteration in levels of chaperone proteins, we analyzed the abundance of $n = 119$ chaperones detected in our proteomics dataset (derived from a published list of $n = 190$ chaperones[49]). Notably, there was no congruent pattern of chaperone abundance; hierarchical clustering showed chaperone sets with either increased, decreased, or no change in abundance across the time-course of disease in the rNLS8 mouse cortex compared to controls (Supplementary Fig. 13). Indeed, some HSP40 family members showed divergent patterns of abundance; for example, in early disease (4 wk) DNAJB5 shows no change (1.14-fold change by immunoblot; Supplementary Fig. 7) but DNAJB6b was significantly decreased (0.6-fold change by immunoblot; $P = 0.003$; Supplementary Fig. 13e–i). Importantly, these data demonstrate that chaperone abundance is regulated through diverse mechanisms in the rNLS8 mouse cortex. Taken together, these data indicate that the abundance of a distinct subset of magenta module chaperones are transiently increased via a post-transcriptional mechanism in rNLS8 mouse cortex neurons specifically in the earliest stages of disease.

## DNAJB5 decreases TDP-43 levels and cytoplasmic TDP-43 aggregation in cells and neurons

Our results revealed a module of protein folding components that may be increased as potential anti-aggregation factors in response to cytoplasmic TDP-43 aggregation in disease. Amongst the magenta module proteins, DNAJB5 showed (i) the highest fold change increase at pre-onset and onset in the quantitative proteomics, (ii) disease-related increases in the cytoplasm of neurons, and (iii) co-localization with TDP-43 puncta in the motor cortex of rNLS8 mice (Fig. 5a). In addition, the relationship between DNAJB5 and cytoplasmic TDP-43

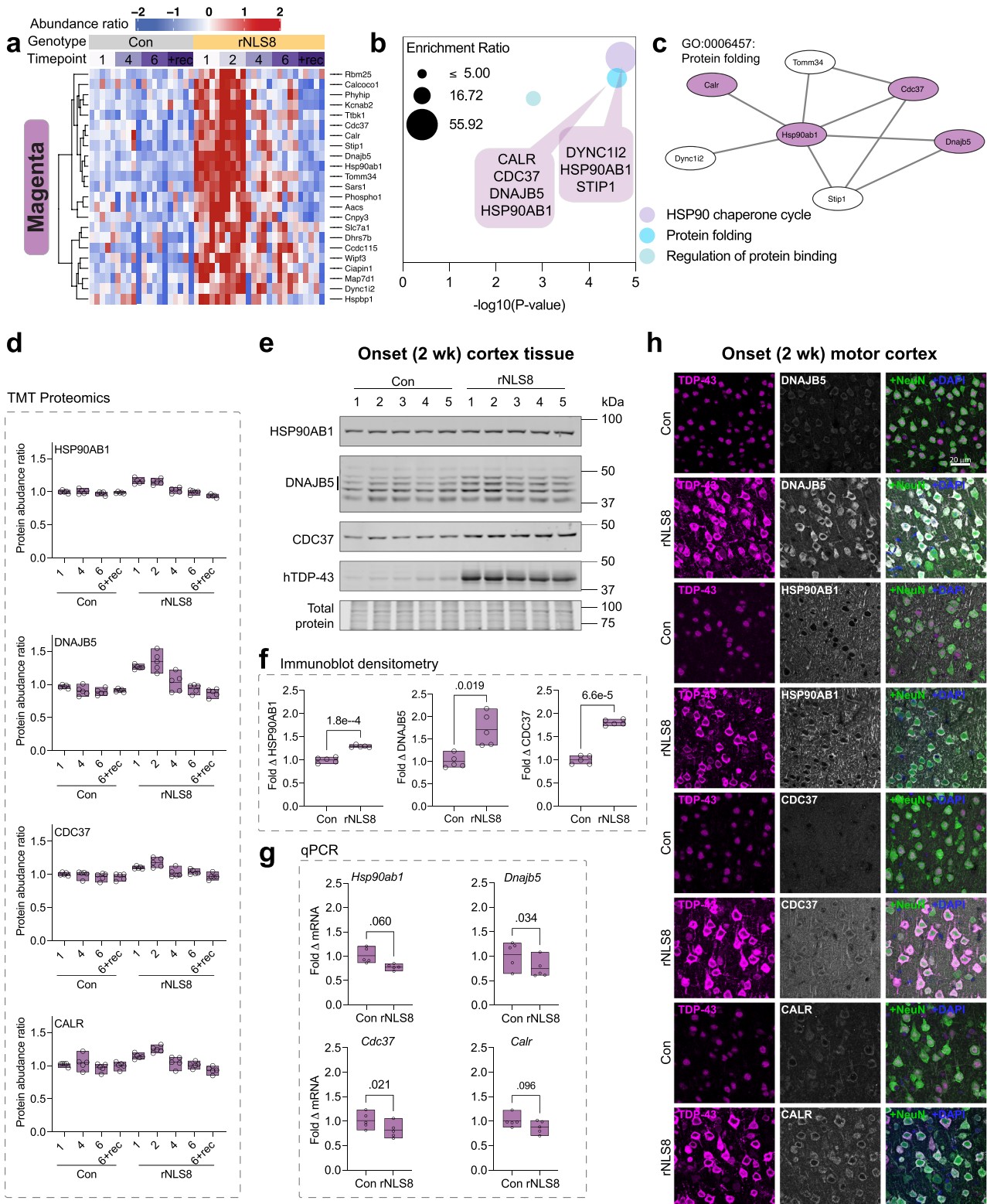

has not previously been explored. We therefore hypothesized that DNAJB5 may be an endogenous protective factor that acts to inhibit TDP-43 aggregation in neurons.

To determine whether increased levels of DNAJB5 protect against TDP-43 aggregation, we first over-expressed DNAJB5 with several cytoplasmic, aggregation-prone, and disease-mimicking TDP-43 variants in HEK293T cells and analyzed changes in TDP-43 solubility and formation of puncta (defined as round TDP-43-positive structures that

are >0.5 μm diameter). TDP-43 wild-type (TDP-43$^{WT}$), cytoplasmic nuclear localization signal mutant (TDP-43$^{\Delta NLS}$), cytoplasmic RNA-recognition motif mutant (TDP-43$^{\Delta NLS/4FL}$), and cytoplasmic mutant (TDP-43$^{\Delta NLS/2KQ}$) that mimics acetylation previously identified in spinal cord tissue of ALS cases[50] with carboxy-terminal tagged monomeric GFP (mGFP) were co-expressed with human DNAJB5-FLAG or inactive control (non-fluorescent mutant EGFP[51]; EGFP$^{Y66L}$-FLAG; herein referred to as CTRL-FLAG).

**Fig. 4 | Protein folding components are specifically increased in abundance in rNLS8 mouse cortex neurons by post-transcriptional mechanisms at the earliest stages of disease. a** Heatmap of protein abundance ratios in control (Con) and rNLS8 mouse cortex identified in the magenta module ($n = 5$ mice per group). **b** Gene ontology analysis of magenta module proteins. Protein sets were analyzed using Metascape[108] where significantly enriched gene ontology terms were identified using the hypergeometric test and Benjamini-Hochberg $P$ value correction algorithm. Plotted by significance ($-\log_{10} P$ value) and overlaid with callouts that list associated proteins. **c** Protein–protein interaction network of protein folding factors in the magenta module. **d** Protein abundance ratio from quantitative tandem mass tag (TMT) proteomics of heat shock protein 90 alpha family class B member 1 (HSP90AB1), DnaJ homolog subfamily B member 5 (DNAJB5), cell division cycle 37 (CDC37) HSP90 co-chaperone, and calreticulin (CALR). **e** Immunoblot of RIPA-soluble whole cortex lysates at disease onset (2 wk). Tissue was probed for HSP90AB1 (90 kDa), DNAJB5 (40–45 kDa, three bands, Supplementary Fig. 8), and CDC37 (50 kDa). **f** Fold change in protein levels relative to the mean of control mice and normalized to total protein loading from immunoblot densitometry. **g** qPCR quantification of fold change in mRNA transcript level of target genes in the rNLS8 cortex at disease onset (2 wk) relative to the mean of control mice and normalized to the *Gapdh* housekeeping gene. Differences in the means of control and rNLS8 mice were determined using a two-sided paired $t$ test, where $P < 0.05$ was considered statistically significant. **h** Immunofluorescence microscopy of the primary motor cortex in control and rNLS8 mice at disease onset. Samples were immunolabeled for each target, DNAJB5, HSP90AB1, CDC37, and CALR (*top to bottom*), and co-labeled for TDP-43, NeuN (pan-neuronal marker) and DAPI. Representative images from $n = 3$ mice. Scale bar = 20 µm. All data are from $n = 5$ mice per group (open circles). Data presented as floating bars in (**d**, **f**, **g**) show a line at the mean and the range displayed by floating bars. Source data are provided as a Source Data file.

DNAJB5 over-expression caused a significant decrease in the soluble and insoluble protein levels of each of the disease-mimicking TDP-43 variants tested compared to controls (Fig. 5b–d and Supplementary Fig. 14). DNAJB5 over-expression did not alter transcript levels of exogenous TDP-43 variants, indicating that the effects observed occur at a protein rather than transcript level (Supplementary Fig. 16). Furthermore, DNAJB5 over-expression significantly decreased the levels of insoluble phosphoTDP-43 (pS403/S404) and phosphoTDP-43(pS409/S410) by 50% and 80%, respectively, in cells expressing TDP-43$^{\Delta NLS/2KQ}$-mGFP (Fig. 5c, d). DNAJB5 over-expression similarly decreased phosphoTDP-43 levels in each of the other cytoplasmic TDP-43 variants tested, suggesting a broad effect against pathology-mimicking variants of TDP-43 (Supplementary Fig. 14). Furthermore, CRISPR knockout of *DNAJB5* in HEK293 Cas9 stable cells over-expressing TDP-43$^{\Delta NLS}$-mGFP or TDP-43$^{\Delta NLS/2KQ}$-mGFP resulted in a significant increase in insoluble TDP-43 and phosphoTDP-43 (Supplementary Figs. 17 and 18), suggesting that endogenous DNAJB5 is important in mediating the pathological states of cytoplasmic TDP-43. Interestingly, DNAJB5 and wild-type TDP-43 co-expression increased phosphoTDP-43 levels in cells, suggesting a differential effect depending on sub-cellular localization of TDP-43.

To determine the effect of DNAJB5 over-expression on the number of puncta per cell, we conducted quantitative single cell analyses. DNAJB5 over-expression resulted in a dramatic decrease in the proportion of cells containing TDP-43$^{\Delta NLS/2KQ}$ puncta (Fig. 5e, f). This was also the case for each of the other TDP-43 isoforms examined in HEK293 cells, whereby co-expression with DNAJB5 resulted in a 60%, 64%, and 68% decrease in the proportion of cells with puncta in TDP-43$^{\Delta NLS/4FL}$, TDP-43$^{\Delta NLS}$, and TDP-43$^{WT}$ samples, respectively (Supplementary Fig. 15). DNAJB5 also significantly decreased the fluorescence intensity and number of puncta formed per cell when co-expressed with each of the TDP-43 variants (Fig. 5e, g, h and Supplementary Fig. 15). Further validation in primary cortical neurons confirmed that DNAJB5 over-expression significantly decreased the number of TDP-43$^{\Delta NLS/2KQ}$-mGFP puncta per neuron and the proportion of cells with puncta (Fig. 5i–k). In parallel experiments, RIPA-soluble TDP-43$^{\Delta NLS/2KQ}$-mGFP co-immunoprecipitated with DNAJB5-FLAG and endogenous HSPA1A (Supplementary Fig. 19). Taken together, these findings demonstrate that DNAJB5 stably binds soluble cytoplasmic TDP-43 to decrease levels of both soluble and insoluble cytoplasmic TDP-43, resulting in a dramatic decrease in the formation of TDP-43 puncta in HEK293 cells and primary cortical neurons.

## Dnajb5 knockout in mice alters the abundance of neurodegeneration-associated proteins

DNAJB5 protein is most highly expressed in cerebral cortex in humans compared to other brain regions[52,53], and in addition *Dnajb5* transcripts are present throughout the brain and in alpha motor neurons of the spinal cord in mice[54,55]. It follows that alterations in DNAJB5 levels may have implications for the central nervous system (CNS). Therefore, we analyzed the cortex proteome of adult *Dnajb5* knockout mice that were previously developed by CRISPR/Cas9[56–58] (herein referred to as *Dnajb5$^{KO/KO}$*; Supplementary Fig. 8). Cortex tissues from *Dnajb5$^{KO/KO}$* mice showed an increase in the abundance of $n = 119$ proteins and decrease in $n = 82$ proteins compared to *Dnajb5$^{WT/WT}$* mice (Fig. 5l). Altered proteins (increased and decreased) were enriched in those involved in regulation of neurogenesis, synaptic biology and function, and RNA localization (Supplementary Data 4). Interestingly, we identified a protein-protein interaction network comprised of altered proteins previously associated with ALS and FTLD including FUS, MAPT (tau), HNRNPA1, HSPA5, EPHA4, and KRAS (Fig. 5m). Of note, *Dnajb5* knockout did not alter the abundance of endogenous TDP-43 or other chaperones in the cortex, except for HSPA5, CHORDC1, and PPIB, which were significantly increased. Combined, these data show that DNAJB5 is an important protein in the cortex for the regulation of levels of proteins associated with ALS, FTLD, neurogenesis, and synaptic function.

## DNAJB5 protects against cytoplasmic TDP-43-associated motor impairments in mice

We next evaluated whether *Dnajb5* knockout could alter cytoplasmic TDP-43-associated motor deficits in vivo. Wild type, heterozygous, and homozygous *Dnajb5* knockout mice were injected with adeno-associated virus (AAV) serotype 9 encoding TDP-43$^{\Delta NLS}$-myc or GFP-myc control driven by the human synapsin1 promoter for neuronal expression in the CNS (Fig. 5n). Bilateral intracerebroventricular injections were conducted at postnatal day 0 and mice were subjected to weekly weighing and neuro-scoring and bi-weekly grip strength and open field testing for 12 weeks. *Dnajb5$^{KO/KO}$* mice were previously reported by the International Mouse Phenotyping Consortium to show mild neurological phenotypes associated with hyperactivity[58]. Indeed, we also found that 12-week-old *Dnajb5$^{KO/KO}$* mice were hyperactive when assessed by open field test ($P = 0.0248$; Supplementary Fig. 20). In addition, over-expression of TDP-43$^{\Delta NLS}$-myc significantly increased hyperactivity of mice in *Dnajb5$^{WT/WT}$*, *Dnajb5$^{KO/WT}$*, and *Dnajb5$^{KO/KO}$* groups ($P < 0.001$ for ambulatory distance; $P = 0.002$ for vertical counts), although there was no additive effect of *Dnajb5* knockout with TDP-43$^{\Delta NLS}$-myc expression on this hyperactivity phenotype (Supplementary Fig. 20).

AAV-mediated neuronal expression of TDP-43$^{\Delta NLS}$-myc in the brain and spinal cord resulted in an acceleration in the time to onset of abnormal hindlimb splay phenotype in all groups ($P = 0.0001$; Fig. 5n). There was a significant effect of *Dnajb5* knockout on motor impairments, and *Dnajb5$^{KO/KO}$* mice expressing TDP-43$^{\Delta NLS}$-myc demonstrated the fastest decline to collapsing hindlimb splay compared to *Dnajb5$^{KO/WT}$* ($P = 0.0351$) and *Dnajb5$^{WT/WT}$* ($P = 0.0269$) mice (Fig. 5n). Similarly, *Dnajb5$^{KO/KO}$* mice expressing TDP-43$^{\Delta NLS}$-myc demonstrated the greatest impairment in grip strength compared to control groups ($P = 0.012$; Supplementary Fig. 21). Mice expressing TDP-43$^{\Delta NLS}$-myc showed no decline in weight or rotarod

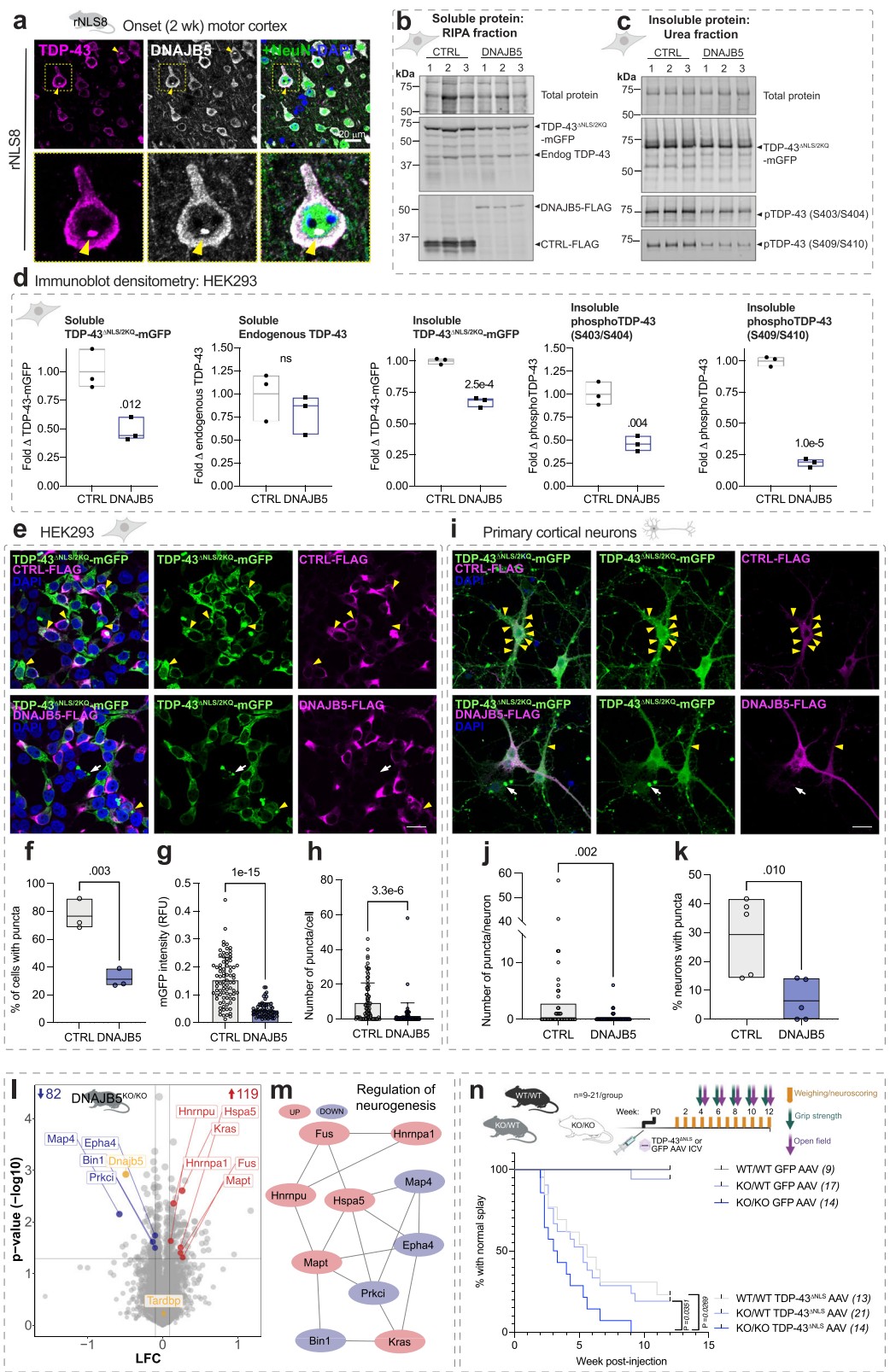

performance compared to GFP-myc controls over 12 weeks, suggestive of an early disease stage in this model (Supplementary Fig. 21). Supporting this, at 12 weeks there was no phosphoTDP-43 detected in the insoluble protein fraction of the cortex in mice expressing TDP-43$^{\Delta NLS}$-myc, despite a high transduction efficiency of AAV9 in the cortex (Supplementary Fig. 22). At 3 months there was no significant difference in cortex soluble or insoluble levels

of TDP-43 between WT, heterozygous, or homozygous *Dnajb5* knockout mice injected with TDP-43$^{\Delta NLS}$-myc (Supplementary Fig. 22), however investigations at later disease stages such as 6 months are warranted. Combined, these data provide in vivo evidence to support the importance of DNAJB5 as a key protective factor against the onset of cytoplasmic TDP-43-associated motor impairments.

**Fig. 5 | Over-expression of DNAJB5 has anti-aggregation activities in cells, and knockout of DNAJB5 exacerbates cytoplasmic TDP-43-associated motor impairments in vivo. a** Primary motor cortex in rNLS8 mice at disease onset. Yellow arrow and zoom showing a TDP-43-positive punctum. Scale bar = 20 μm. Immunoblot analysis of (**b**) soluble and (**c**) insoluble fractionated lysates. **d** Immunoblot densitometry analysis of soluble TDP-43$^{ΔNLS/2KQ}$-mGFP, soluble endogenous TDP-43, and insoluble TDP-43$^{ΔNLS/2KQ}$-mGFP, phosphoTDP-43(S403/ S404), and phosphoTDP-43(S409/S410) levels. **e** HEK293 cells immunolabeled with anti-FLAG. Scale bar = 20 μm. Yellow arrows = inclusions in co-transfected cells and white arrows = TDP-43$^{ΔNLS/2KQ}$-mGFP only. **f** Proportion of cells with puncta. **g** TDP-43$^{ΔNLS/2KQ}$-mGFP relative fluorescent units (RFU) per cell. **h** Number of puncta per cell. Data in (**g**, **h**) are from $n = 83$ CTRL and $n = 69$ DNAJB5 cells, error bars represent standard deviation. Data from (**a–h**) are from $n = 3$ independent experiments. **i** Primary cortical neurons expressing CTRL-FLAG or DNAJB5-FLAG with TDP-43$^{ΔNLS/2KQ}$-mGFP and immunolabeled with anti-FLAG. Scale bar = 20 μm. **j** Number of puncta formed per neuron from $n = 62$ CTRL and $n = 123$ DNAJB5 neurons.

**k** Proportion of neurons with puncta. Data in (**i**, **j**, **k**) are from $n = 5$ independent experiments. Data points in (**g**, **h**, **j**) show single-cell data from one representative independent experiment. Data presented as floating bars in (**d**, **f**, **k**) show a line at the mean and the range. Differences between the means for (**d**, **f**, **k**) were determined using two-sided unpaired $t$ tests and (**g**, **h**, **j**) were determined using two-sided $t$ test with Welch's correction. **l** Volcano plots of TMT quantitative proteomics from *Dnajb5* knockout mouse cortex, log fold change (LFC; *Dnajb5*$^{KO/KO}$/*Dnajb5*$^{WT/WT}$) and $P$ value (-log10). Inset numbers = significantly decreased (blue) and increased (red; $P < 0.05$, fold change > 1.2) proteins. Data are the means of $n = 3$ mice per group. Statistically significant differences in relative protein abundance was determined using two-sided student $t$ tests. **m** Protein–protein interaction network of significantly altered proteins in *Dnajb5*$^{KO/KO}$ cortex. **n** Kaplan–Meier curves showing the proportion of animals with normal hindlimb splay ($n$ = italicized numbers) over time. Differences in the curves were determined by log-rank (Mantel-Cox) test. Source data are provided as a Source Data file.

## Table 2 | Demographics and clinical features of human brain donors

| Case type | Case code | Diagnosis | Sex | AAO (y) | AAD (y) | Site of Onset | Brain weight (g) | PMD (h) |
|---|---|---|---|---|---|---|---|---|
| Neurologically normal control | H215 | NNDC | F | N/A | 67 | N/A | 1232 | 23.5 |
| Neurologically normal control | H230 | NNDC | F | N/A | 57 | N/A | 1243 | 32 |
| Neurologically normal control | H247 | NNDC | M | N/A | 51 | N/A | 1671 | 31 |
| Case | MN13 | ALS | M | 55 | 55 | Spinal – UL | 1384 | 10 |
| Case | MN15 | ALS + FTD | F | 52 | 54 | - | - | 18 |
| Case | MN22 | ALS | F | 58 | 65 | Spinal – UL | 1138 | 9 |

Unknown, *AAD* age at death, *AAO* age at onset, *N/A* not applicable, *NNDC* non-neurologically diseased control, *PMD* post-mortem delay, *UL* upper limb.

## DNAJB5 distribution is altered in neurons with TDP-43 pathology in human autopsy tissues

We next sought to determine the levels and cellular localization of DNAJB5 in the human motor cortex of non-neurological disease controls and ALS and ALS + FTD cases (Table 2). In post-mortem human brain tissue, DNAJB5 was expressed at a low level in neurons of the motor cortex with no apparent difference in levels between controls and disease cases (Fig. 6). However, in cases with ALS or ALS + FTD, DNAJB5 appeared enriched perinuclearly in neurons containing perinuclear phosphoTDP-43 aggregates (Fig. 6), suggesting sequestration of DNAJB5 with TDP-43 inclusions.

## Protein changes in rNLS8 mice reflect alterations in human TDP-43 proteinopathies

Given the difficulty in confirming alterations in protein levels occurring specifically in the earliest stages of disease in rNLS8 mice also occur at similar time points in human disease, we instead hypothesized that the proteomic signatures of the rNLS8 mouse cortex at late disease would closely correlate with changes seen in end-stage human TDP-43 proteinopathies tissues. We therefore compared the rNLS8 mouse datasets to two recent high-quality human datasets: (1) proteomics from post-mortem frontal cortical tissues from clinically defined (sporadic) ALS, ALS/FTLD, FTLD-TDP and control cases[59], and (2) transcriptomics from post-mortem frontal cortex, temporal cortex, and cerebellum from FTLD-TDP and control cases[60]. We also developed a web application, TDP-map, for targeted meta-analysis of these three datasets, allowing investigation of any proteins of interest across the studies (https://shiny.rcc.uq. edu.au/TDP-map/). We first compared significantly altered proteins from the rNLS8 mouse cortex in late disease (6 wk) with human datasets of TDP-43 proteinopathies (Fig. 7). Of $n = 622$ altered proteins in late-disease mouse cortex, there were $n = 93$ proteins in common with the human proteomic dataset, $n = 166$ genes in common with the human transcriptomic dataset, and $n = 51$ in common across all three datasets (Fig. 7a).

Interestingly, proteins that were decreased in the late disease rNLS8 cortex also showed a trend towards a decrease or down-regulation in the human datasets (Fig. 7b). In addition, proteins that were increased in late disease in the rNLS8 cortex showed a trend for increased protein abundance or transcript up-regulation in the human datasets (Fig. 7c). Regarding the human proteomics datasets[59], the highest degree of concordance was observed in the proteomic signatures of late disease rNLS8 cortex and frontal cortex of the human FTLD-TDP samples, with low overlap with the human ALS and ALS/ FTLD frontal cortex (Fig. 7b, c). This may indicate that the changes in the rNLS8 cortex are particularly relevant for identifying the molecular mechanisms of FTLD-TDP. Gene ontology analysis of the $n = 93$ proteins in common between the cortex of rNLS8 mice and FTLD-TDP humans showed a shared depletion of proteins involved in chemical synaptic transmission and synaptic signaling and an enrichment in response to wounding and neutrophil degranulation (Fig. 7d). Common proteins that were decreased in late disease in rNLS8 cortex and human FTLD-TDP revealed a protein-protein interaction network comprising calcium ion import proteins and another comprising mitochondrial respiratory chain complex I components (Fig. 7e). Interestingly, several of these components were part of a larger mitochondrial respiratory chain complex I protein-protein interaction network that was significantly increased in mice prior to disease onset (1 wk; Fig. 2f).

To further compare the biological processes altered in disease between mouse and human, the WCNA modules from the longitudinal rNLS8 mouse cortex proteomics analysis were compared with modules generated from proteomics of human autopsy samples[59] (Fig. 7f). Notably, each of the WCNA modules from the rNLS8 cortex demonstrated concordance with at least one human disease module, highlighting relevant overlapping biochemical signatures in TDP-43-driven disease. For example, both studies identified decreased abundance of proteins involved in chemical synaptic transmission in late disease [module 4 turquoise (rNLS8 mice) = M1 turquoise (human)]. However, analysis of the rNLS8 cortex also revealed that the loss of turquoise

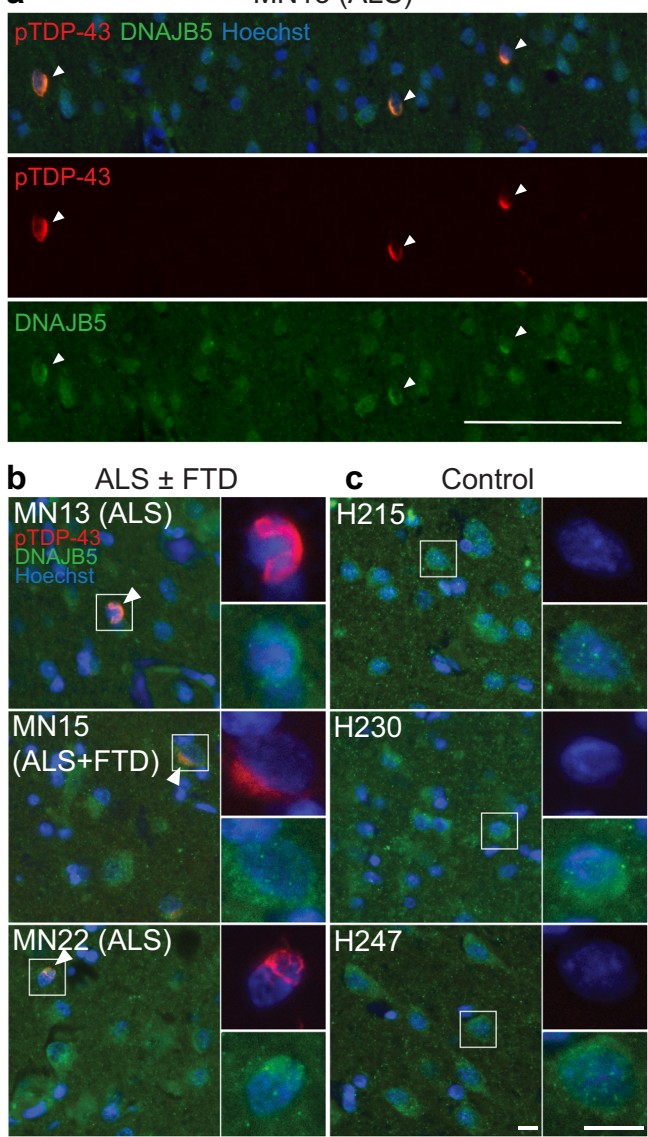

**Fig. 6 | Localization of DNAJB5 and phosphoTDP-43 in the post-mortem motor cortex of human ALS and ALS-FTD cases.** Immunohistochemical labeling of DNAJB5 (green) and phosphoTDP-43 (red) in the post-mortem motor cortex of $n = 3$ non-neurologically diseased controls and $n = 3$ ALS or ALS + FTD cases (Table 2). **a** Wide field of view of MN13 (ALS). White arrows show neurons with co-localization of phosphoTDP-43 and DNAJB5. Scale bar = 100 μm. **b** Zoom of ALS and ALS-FTD cases MN13 (ALS), MN15 (ALS + FTD) and MN22 (ALS); and (**c**) neurologically normal cases H215, H230 and H247. Co-localization indicated by white arrows. Nuclei counterstained with Hoechst (blue). Scale bars, 10 μm. Abbreviations: ALS, amyotrophic lateral sclerosis; FTD, frontotemporal dementia.

proteins was initiated prior to disease onset. In addition, the magenta module, which comprised anti-aggregation protein folding factors, showed the greatest intersection with the M8 pink cell differentiation human module (albeit with only two proteins in common, CDC37 and MAP7D1)[59]. However, proteins in this M8 pink module (human) were decreased in end-stage FTLD-TDP, whereas magenta module (rNLS8 mice) proteins were increased in pre-onset mice (from 1 week off dox; Supplementary Fig. 23) and this was the only module-module comparison showing the opposite direction of change between mouse and human. Our independent gene ontology analysis of the M8 pink (human) module showed a depletion of proteins that regulate the HSF1-mediated heat shock response and protein

stabilization in end-stage FTLD-TDP disease (Supplementary Data 4). Therefore, the rNLS8 mouse longitudinal proteomics datasets provide insights into key groups of proteins that drive disease initiation and those that may be increased to compensate for pathological TDP-43 accumulation.

## Discussion

Large-scale human autopsy studies have revealed numerous disease-related changes that contribute to TDP-43 proteinopathies[59,60]. However, there is growing evidence that the biological drivers of disease initiation and progression likely change across the disease course, and the important mechanisms in early disease may not be reflected in end-stage tissue[19,39,40,61]. We report a comprehensive longitudinal quantitative proteomics analysis of cortex tissue from the validated rNLS8 TDP-43 mouse model of ALS and FTLD. We identified distinct and dynamic biochemical signatures of TDP-43-mediated disease, including very early alterations in proteins involved in neuronal function (Fig. 8). WCNA identified modules of proteins with correlated abundance throughout disease, highlighting dysregulation of biological processes such as protein transport, myelination, metabolic processes, and chemical neurotransmission at different stages of disease. Importantly, we also identified a group of proteins enriched in protein folding factors that demonstrated an early but transient increase in protein abundance in rNLS8 mice. Over-expression of the protein folding factor, DNAJB5, in HEK293 and primary cortical neuronal cultures decreased levels of pathological and insoluble TDP-43. In addition, *Dnajb5* knockout exacerbated hindlimb and forelimb motor impairments in mice expressing cytoplasmic TDP-43 in the CNS. This group of protein folding factors, including DNAJB5, may therefore represent a protective mechanism that is initiated early but is not maintained throughout disease. Meta-analysis of the rNLS8 cortex proteomics dataset with human brain datasets revealed many commonalities between late-disease mouse and end-stage human TDP-43-associated disease, indicating strong validity of this mouse model. Thus, the altered biological processes identified at early disease timepoints in the rNLS8 mice may provide insights of relevance to the earliest stages of human TDP-43-mediated disease pathogenesis that would not otherwise be detected due to the inaccessibility of human brain and spinal cord tissues.

Our work revealed unexpected patterns of activation of potentially protective biological processes in the rNLS8 mouse cortex, in addition to previously identified signatures of glial activation and neuron loss associated with neurodegeneration. The presence of distinct changes in protein abundance from early to late disease has been well-established in large-scale proteomic analyses of human Alzheimer's disease brains[61-63], but has not been clearly defined for TDP-43 proteinopathies. Previous studies have shown upregulation of the integrated stress response and anti-apoptotic factors in the rNLS8 mouse cortex prior to disease onset[19], and a decrease in mitochondrial respiratory chain complex proteins in the sciatic nerve axoplasm in early disease[40]. Our study has now advanced these findings and defined a comprehensive proteomic signature of the cortex at each stage of TDP-43-mediated disease. For example, decreased levels of proteins mediating neuron projection organization (proteins that mediate assembly, arrangement, and disassembly of neuronal processes), and increased levels of proteins related to protein quality control, macroautophagy and chaperones, define early disease responses to cytoplasmic TDP-43. In contrast, late disease is characterized by accelerating decreases in proteins involved in chemical synaptic transmission and increases in neuroinflammatory processes, reflecting advanced stages of neurodegeneration.

Our longitudinal analysis allowed deconvolution of cell-type-specific contributions to disease signatures over time. We demonstrated cell-autonomous responses of neurons (turquoise module), which was defined by neuronal system proteins and chemical synaptic

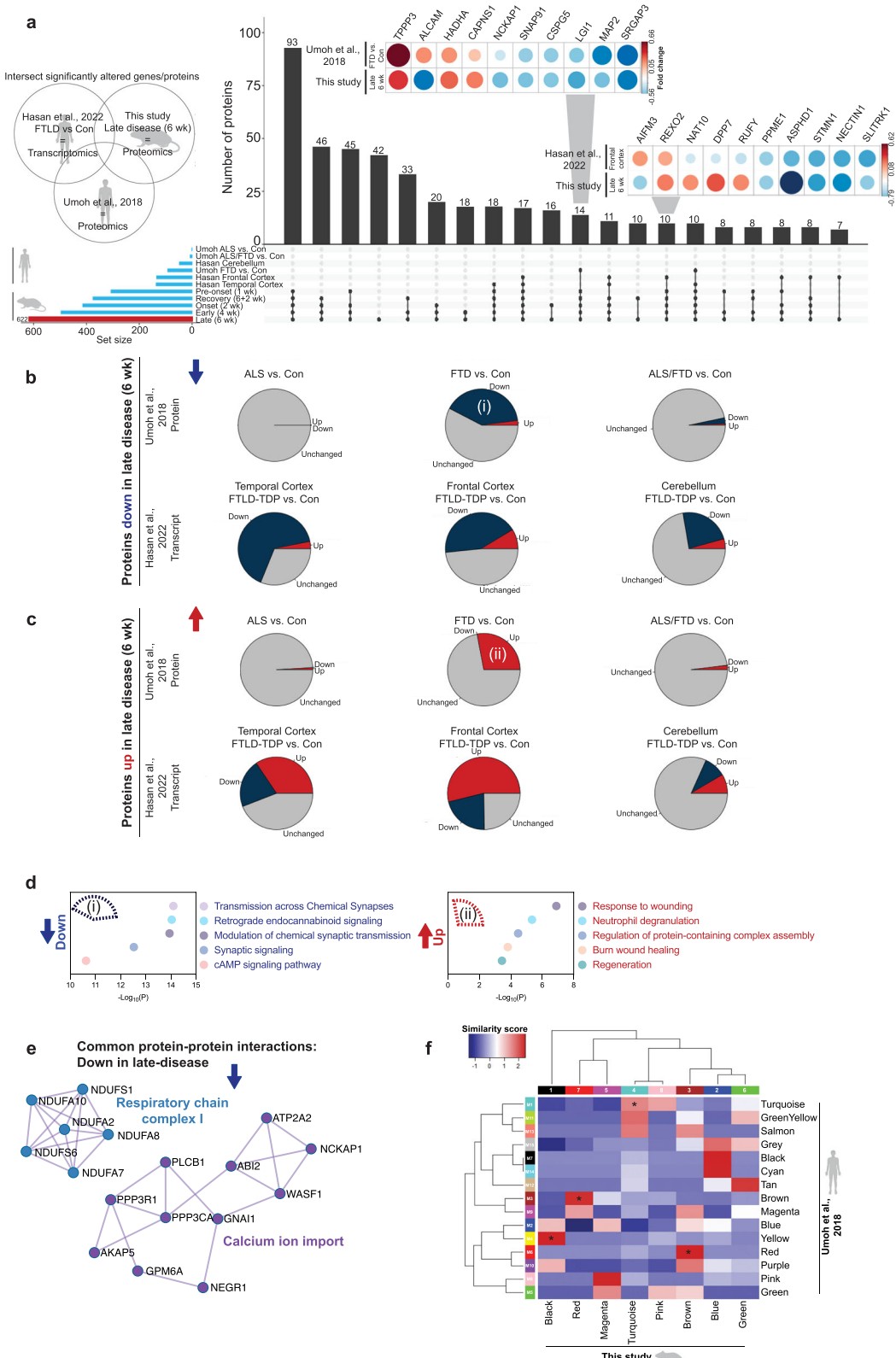

transmission. Previous work has shown that depletion in proteins involved in chemical synaptic transmission is a feature of end-stage disease[59]. However, a decrease in the abundance of these proteins was initiated prior to disease onset in rNLS8 mice and partially returned to control levels in recovery, suggesting a greater than expected degree of plasticity of neuronal changes in disease. A step-wise decline in synaptic proteins is also a feature of asymptomatic and symptomatic

Alzheimer's disease[62]. This could result from a loss of TDP-43 function in the rNLS8 mouse cortex, as knockdown of *Tardbp* has been shown to result in synaptic deficits[64] and mis-splicing of synaptic proteins[30,31]. In addition, the non-cell-autonomous contribution of oligodendrocytes and microglia in disease was further revealed. Enrichment of oligodendrocyte proteins (black module) is suggestive of changes in neuronal ensheathment in late disease as well as in recovery.

**Fig. 7 | Substantial overlap of altered proteins in the cortex of late-disease rNLS8 mice (6 wk) and end-stage human post-mortem ALS, ALS/FTLD, and FTLD-TDP cases.** Meta-analysis of significantly altered proteins in late disease rNLS8 mice (6 wk) with selected published datasets of end-stage human TDP-43 proteinopathies. **a** Upset plot representing the intersection of proteins that are altered in abundance (increased and decreased) in three studies, (1) this study, (2) human end-stage ALS, FTLD or ALS/FTLD proteomics[59], and (3) human end-stage transcriptomics of FTLD cerebellum, temporal cortex, and frontal cortex[60]. Horizontal bars represent the total number of altered proteins/transcripts (set size) in each sample. Vertical bars represent the number of intersecting proteins of each sample comparison (comparisons are denoted as dots connected by lines below the X axis). *Inset:* Top 10 proteins significantly decreased or increased in late disease (6 wk) rNLS8 mouse proteomics and end-stage human FTLD-TDP-43 frontal cortex proteomics[59]. This plot shows the reported fold change. Blue = decreased levels and red = increased levels. Direction of fold change of intersecting proteins that are (**b**)

decreased and (**c**) increased in late-disease (6 wk) rNLS8 mice, in human proteomic[59], and corresponding genes in human transcriptomic[60] datasets. Gray = unchanged, blue = down, and red = up. **d** Gene ontology and (**e**) protein-protein interactions of significantly decreased or increased proteins in late-disease (6 wk) rNLS8 mouse proteomics and end-stage human FTLD-TDP-43 frontal cortex proteomics[59]. Protein sets were analyzed using Metascape[108] where significantly enriched gene ontology terms were identified using the hypergeometric test and Benjamini-Hochberg *P* value correction algorithm. **f** Comparison of WCNA modules in this study with human frontal cortex proteomics from clinically defined ALS, FTLD, and ALS/FTLD cases[59]. Blue is low and red is high percentage similarity between protein lists in modules. Modules that were significantly enriched (*) of proteins (compared to the background list) were identified using a two-sided Fisher's exact test with multiple testing correction (Holm correction). Black(1):Yellow(M4) $P = 1.78 \times 10^{-19}$, Red(7):- Brown(M3) $P = 1.11 \times 10^{-19}$, Turquoise(4):Turquoise(M1) $P = 2.21 \times 10^{-18}$, and Brown(3):Red(M6) $P = 4.28 \times 10^{-7}$. Source data are provided as a Source Data file.

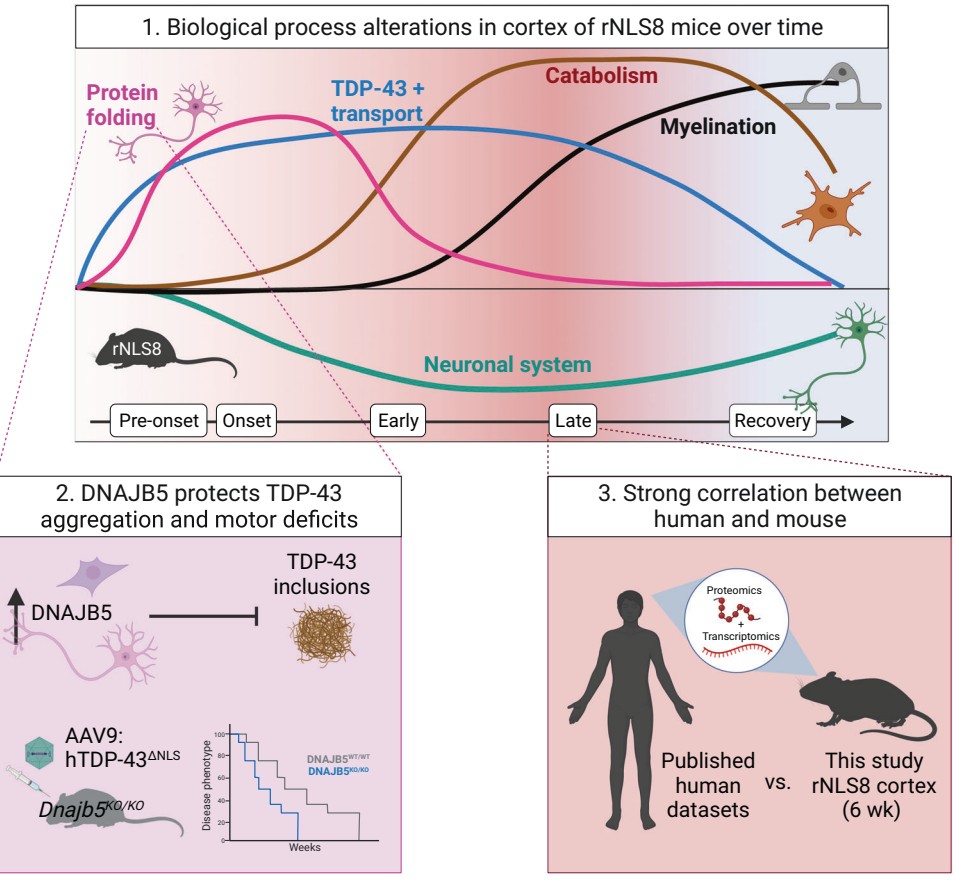

**Fig. 8 | Cytoplasmic TDP-43-mediated disease is defined by dynamic alterations to the cortex proteome over time.** 1. A schematic of the alterations to biological processes in the rNLS8 cortex over the time course of disease identified by quantitative proteomics. Increases in the abundance of protein folding factors and transport components define early disease, and catabolism and myelination define late disease molecular signatures. Accelerating decreases in neuronal system and chemical synaptic transmission components occurs throughout the disease trajectory with a partial reversal of some proteins to control levels in recovery. 2. One of the increased protein folding factors identified in the rNLS8 cortex, DNAJB5,

showed anti-aggregation activities in cell and neuronal culture models. Further, *Dnajb5* knockout mice demonstrated worsened motor impairments in an AAV model of cytoplasmic TDP-43 disease. This shows that DNAJB5 protects against cytoplasmic TDP-43 aggregation and motor deficits. 3. A strong correlation in the proteomic signature of late disease mouse and post-mortem cortex tissue of cases with TDP-43 proteinopathy validate the rNLS8 mouse model. Therefore, findings at early disease timepoints in the rNLS8 cortex may accelerate our understanding of the mechanisms driving disease onset and early disease.

Interestingly, TDP-43 may be required for *Mbp*, *Mog*, and *Plp1* transcript expression[65,66], and oligodendrocyte-specific deletion of *Tardbp* in mice results in a decrease in these mRNAs[66,67]. Indeed, decreased MBP protein and myelin sheath defects are characteristic of sporadic ALS motor cortex and spinal cord[68]. Future studies should investigate whether this increase in oligodendrocyte proteins is protective and coincides with increased myelination in late disease and recovery in

the rNLS8 mouse cortex. Lastly, increased microglial proteins were associated with the brown module and showed an increased abundance of lysosomal proteins. This is consistent with findings from transcriptomics of microglia isolated from rNLS8 mouse cortex in early disease[39], and findings that phagocytic microglia progressively become dysfunctional in the motor cortex in rNLS8 mice and human ALS cases[21].

A notable finding from our work is the increase in protein folding factors that occurs prior to onset of motor impairments and TDP-43 inclusion formation in the rNLS8 mice. Motor neurons are considered to be poor at inducing chaperone expression under stress, due to their inability to activate a heat shock transcription factor 1-mediated heat shock response or to up-regulate HSP70 expression[69]. However, here we show that, amongst such protein folding factors, DNAJB5, HSP90AB1, CALR, and CDC37 all exhibit increased levels in the cytoplasm of hTDP-43$^{\Delta NLS}$-positive neurons in the primary motor cortex of rNLS8 mice at disease onset. Further, we demonstrate that this increase in abundance is mediated by currently uncharacterized post-transcriptional mechanisms (possibly via non-coding RNAs[70] or RBPs[71]), given that there was no increase in transcript level to correlate with the heightened protein abundance. Previous studies have also reported a divergence of protein levels from RNA levels of some HSPs, with increases in HSPB1 in SOD1$^{G93A}$ mice being accompanied by no change in *Hspb1* transcripts[72]. These findings suggest that uncharacterized post-transcriptional mechanisms drive rapid increases in the abundance of defined subsets of chaperones in the cortex in disease. Further, in another study, proteomics of 1,000 Alzheimer's disease brains revealed disease-specific changes that were not observed in transcriptomic data from the same cases[73]. This is likely a brain-specific feature given that a quantitative proteome map of the human body revealed that brain tissue shows the lowest correlation between RNA and protein levels compared to all other tissues[74]. In fact, the brain exhibits the largest number of genes that are enriched only at the protein level[74]. Together, these findings indicate that post-transcriptional mechanisms regulate many protein alterations in the cortex.

The increased abundance of protein folding factors likely confers anti-aggregation properties to neurons in the cortex. Indeed, HSP90 and CDC37[75], and HSP90 and STIP1[76] interact with TDP-43 to mediate its solubility and clearance through the autophagy-lysosome pathway in cells[77]. We demonstrated that DNAJB5, which exhibited one of the highest fold increases in abundance in our proteomics dataset at disease onset, co-localized with TDP-43 inclusions in rNLS8 mice and in human post-mortem ALS motor cortex and had potent anti-aggregation effects against cytoplasmic TDP-43 variants. Interestingly, DNAJB5 may act specifically on cytoplasmic TDP-43, since over-expression of DNAJB5 with wildtype TDP-43 had no effect on levels of insoluble TDP-43[16]. Further, over-expression of DNAJB5 with disease mutant FUS$^{R521H}$ increased inclusions[78], and had no or little effect on polyglutamine expanded huntingtin[78,79] or ataxin-3[80] insoluble protein levels. Other HSP40 family members may also mediate TDP-43 aggregation, since overexpression of DNAJB1[17] and DNAJB2a[16] significantly decreased insoluble TDP-43 and phosphoTDP-43 in cells. Future work should investigate the interactions between cytoplasmic TDP-43 and DNAJB5 and the mechanisms mediating DNAJB5 anti-aggregation activity. Together, these findings suggest protein folding factors are likely increased in abundance in the rNLS8 cortex in response to cytoplasmic TDP-43 accumulation in neurons to combat protein aggregation, although further analysis of the potential effects of the concurrent loss of function of TDP-43 on this pathway is warranted.

Our results also suggest a potential role for DNAJB5 in maintaining synaptic biology in the cortex. We show that *Dnajb5* knockout significantly altered the levels of proteins involved in synaptic translation, structure, signaling, long term synaptic potentiation, and learning and memory. In related work, DNAJB5 was identified as a component of a synaptic protein complex, along with 11 sodium and calcium voltage-gated ion channels, in BraInMap, a global proteomic investigation of 1000 multi-protein complexes in the brain[81]. *Dnajb5* transcripts are also upregulated in the striatum of rodents treated with psychostimulants MDMA[82], and D-amphetamine[83], and L-DOPA[84]. Further, *Dnajb5* transcripts are significantly increased in hippocampus of an epileptic mouse model with hyperexcitability[85], a mouse model of

moderate glutamate hyperactivity[86], and pentylenetetrazole-induced excitotoxicty[87] in published microarray and RNAseq datasets. Indeed, cortical hyperexcitability is strongly associated with cytoplasmic TDP-43 in mouse models of ALS[12,88] and in human disease[89]. We hypothesize that, in addition to its anti-aggregation activities, DNAJB5 may protect synapses in hyperexcitable contexts, which may in part explain the worsening of motor impairments in *Dnajb5* knockout mice with cytoplasmic TDP-43 expression in the cortex in this study. It follows that future research could evaluate whether the delivery of exogenous DNAJB5 rescues motor impairments and shortened survival associated with cytoplasmic TDP-43 pathology.

We found that the proteomic signature of the rNLS8 mouse cortex in late disease correlates strongly with that of post-mortem brain tissue from TDP-43 proteinopathies[59,60]. Depletion of chemical synaptic transmission, enrichment of response to wound healing, and neutrophil degranulation were the top biological processes in common with late disease rNLS8 and end-stage human FTLD cortex. However, different initiating etiologies of disease may converge on neuroinflammation and neurodegeneration in the cortex and therefore future studies that validate the relevance of these findings for human TDP-43 proteinopathies is warranted. To facilitate this, we developed a new web-tool, TDP-map. This tool will enable the research community to easily interrogate the patterns of change of any protein or groups of proteins of interest across both our longitudinal rNLS8 mouse cortex proteomics dataset and publicly available human proteomics and transcriptomics datasets (https://shiny.rcc.edu.au/TDP-map/). Demonstrating the utility of this tool, we show correlated and divergent patterns of protein/gene change and highlight key common targets that are altered in both mouse and human disease (Fig. 7).

TDP-43 pathology is now implicated in many neurological diseases, including Alzheimer's disease[90,91], limbic-predominant age-related TDP-43 encephalopathy (LATE)[92,93], Huntington's disease[94], Parkinson's disease[95], supranuclear palsy and corticobasal degeneration[96,97], and chronic traumatic encephalopathy[98]. A greater knowledge base of the cellular and molecular signatures of TDP-43 proteinopathies will facilitate a better understanding of the pathogenesis and strategies for treatment of these diseases. This study complements several proteomics and transcriptomics datasets from human post-mortem end-stage brain tissues of ALS and FTLD cases and provides a longitudinal map of the cortex proteome throughout TDP-43-driven disease in a gold-standard mouse model. The identification of dynamic time-dependent subsets of proteins highlights that cytoplasmic TDP-43-driven disease is complex, multi-factorial, and elicits widespread protein changes from the very earliest disease stages. This approach paves the way for future research investigating other ALS-relevant tissues such as spinal cord, brainstem, and muscle. Modulation of TDP-43 pathology by targeting the distinct set of protein folding factors discovered in this research is a promising avenue for the future development of therapies to treat TDP-43 proteinopathies.

## Methods

### Experimental models

**Ethics statement.** All experiments were carried out in accordance with the Australian Code of Practice for the Care and Use of Animals for Scientific Purposes (8th Edition, 2013). Mice were both bred and housed for all subsequent in vivo mouse studies in a Specified Pathogen-Free (SPF) animal facility with a 12 h light/dark cycle (lights on at 06:00 h) and the room temperature and humidity maintained at $21 \pm 1\,°C$ and $55 \pm 5\%$, respectively. Experiments were conducted with approval from the Animal Ethics Committees of Macquarie University (#2016-026) and The University of Queensland (QBI/131/18 and 2021-AE000451).

Regarding the use of human post-mortem tissue, experiments were conducted with informed donor consent, with approval from the NZ Health and Disability Ethics Committee (14/NTA/208) and The University of Auckland Human Participants Ethics Committee.

**Transgenic TDP-43ΔNLS mouse model of ALS**. TDP-43ΔNLS mice used for TMT10 quantitative proteomics and immunoblotting, referred to as regulatable NLS (rNLS8) mice, were produced from the intercross of hemizygous *tetO*-hTDP-43-ΔNLS line 4 (RRID:IMSR_JAX:014650) mice with hemizygous *NEFH*-tTA line 8 (RRID:IMSR_JAX:025397) on a mixed B6/C3H F1 background, as described previously[18], at Macquarie University.

TDP-43ΔNLS mice used for qPCR, were produced from the intercross of homozygous *tetO*-hTDP-43-ΔNLS mice with hemizygous *NEFH*-tTA mice on a pure C57BL/6JAusb background following >10 generations of backcrossing, at The University of Queensland.

All mice were fed with dox-containing chow (200 mg/kg, Specialty Feeds, Australia). On the removal of dox at a mean age of 10 weeks, expression of hTDP-43ΔNLS was induced, and the mice demonstrated an ALS/FTLD disease phenotype reminiscent of human disease, as described previously[18]. All mice were group housed.

**Transgenic TDP-43Q331K mouse model of ALS**. Transgenic TDP-43Q331K mice (RRID:IMSR_JAX:017933) were bred and maintained as heterozygotes on a C57BL/6NJ background. Transgenic mice (TDP-43Q331K) and litter-matched non-transgenic mice (WT) were used in this study.

**CRISPR/Cas9 Dnajb5 knockout mice**. The C57BL/6NJ-*Dnajb5*em1(IMPC)J/Mmjax (RRID:MMRRC_043738-JAX) mouse strain was generated by CRISPR/Cas9-mediated *Dnajb5* knockout by the Knockout Mouse Project and obtained from Jackson Laboratories[57]. *Dnajb5*KO/KO mice were crossed with C57BL/6JAusB mice to obtain heterozygous *Dnajb5*KO/WT mice for breeding colonies. For experiments, *Dnajb5*KO/WT male and females were time-mated to produce homozygous knockout *Dnajb5*KO/KO, heterozygous *Dnajb5*KO/WT, and wildtype *Dnajb5*WT/WT mice.

**Mouse monitoring**. On the removal of dox, rNLS8 mice were weighed and scored three times per week using our neurological monitoring sheet to determine visible signs of disease onset and monitor disease progression. Onset was defined when hindlimb clasping was observed for two consecutive sessions.

**Cell lines**. HEK293T cells (RRID:CVCL_1926, female origin) were obtained from the American Type Culture Collection and cultured in DMEM/F12 (Gibco) supplemented with 10% (v/v) fetal calf serum (Gibco). HEK293 cells were maintained at 37 °C and 5% $CO_2$ levels in an IncuSafe MCO-170AICUV-PE $CO_2$ Incubator (PHCbi) and passaged every 2 days or at 80% confluency. Routine *Mycoplasma* testing was conducted on actively growing cells.

**Cortical neuronal cultures**. Time-mated female C57BL/6J mice were euthanized 15 days after time mating. Mouse embryos were recovered from mothers and then transferred into cold Hibernate E medium (Gibco, A1247601) + 1% penicillin-streptomycin (Gibco, 15140122). Brains were micro-dissected from the embryos, the meninges removed, and the cortex isolated. Cortex tissue from up to 8 embryos was pooled into 1.5 mL microfuge tubes with Hank's balanced salt solution (HBSS) (Gibco, 14170-112), rinsed twice with fresh HBSS, then resuspended into 270 μL of HBSS. The cortex tissue was digested with 0.25% (v/v) trypsin (Gibco, 15090046) and incubated at 37 °C for 15 min. Trypsinization was stopped with addition of horse serum (Gibco, 26050070) to final concentration 30% (v/v) and DNA digested with 0.1% (w/v) DNase I (Sigma Aldrich, DN25) with a 37 °C incubation for 10 min followed by mechanical trituration to dissociate cells. The cell homogenate was centrifuged at room temperature for 7 min at 1500 rpm. The cell pellet was resuspended in neuronal plating medium, Neurobasal Medium (Gibco, 21103-049) containing 10% (v/v) fetal bovine serum (Gibco, 10099141), 2 mM L-glutamine (Gibco, 25030081), and 1% v/v penicillin/streptomycin, prepared, and equilibrated overnight at 37 °C in a $CO_2$ incubator. Cell counts were performed, and neurons plated in neuronal plating medium onto 24-well plates ($5 \times 10^4$ cells/well) precoated with poly-L-lysine hydrobromide (Sigma, P2636; 1 mg/mL for glass coverslips or 0.1 mg/mL for plasticware) and cultured for 4 h at 37 °C in a $CO_2$ incubator, to allow cell attachment. At 4 h post-plating, the medium was changed to neuronal maintenance medium, Neurobasal Medium (Gibco, 21103-049) containing 2% (v/v) B27 supplement (Gibco, 17504-044), 2 mM L-glutamine (Gibco, 25030081), and 1% v/v penicillin/streptomycin) for all subsequent culture maintenance and transduction experiments.

## Method details

**Recombinant AAV preparations**. All AAV preparations were produced by transient transfection of adherent human embryonic kidney (HEK) 293 T cells (ATCC, CRL-3216) with three plasmids: (1) pAd5 helper plasmids encoding adenoviral proteins required for AAV replication, (2) pR2C9 plasmid encoding REP of AAV2 and CAP of AAV9, and (3) gene transfer plasmid containing the expression cassette between AAV2 ITR elements. The transfection was performed using linear polyethyleneimine hydrochloride (40,000, PEI MAX, Polysciences, 49553-93-7). For each vector $5 \times 15$ cm dishes were transfected and subsequently AAV particles were purified from cell pellet and media using iodixanol gradient purification as previously reported[99].

**AAV-mediated TDP-43ΔNLS over-expression in Dnajb5 knockout mice**. Over-expression of TDP-43ΔNLS-myc or EGFP-myc in the CNS was achieved via adeno-associated viruses AAV9-hSyn1-TDP-43ΔNLS-myc and AAV9-hSyn1-EGFP-myc produced at the Vector and Genome Engineering Facility of the Children's Medical Research Institute (Australia). Postnatal day 0 mice were cryo-anaesthetized prior to bilateral intracerebroventricular stereotaxic injections 2 mm either side of the midpoint between lambda and bregma to 2.5 mm depth using a 30 G needle connected to a Hamilton syringe via polyethylene tubing. A total viral titer of $2 \times 10^{10}$ viral genome copies was delivered in 4 μL (2 μL per hemisphere). Group sizes of 9–21 were used and experimenters were blinded to genotype and AAV treatment.

**Behavioral testing**. Weighing and hindlimb reflex scoring was conducted weekly. The hindlimb reflex was scored on a scale of 4 to 0 based on hindlimb reflex extension (or splay) and the onset of hindlimb clasping. Normal hindlimb reflex without clasping, 4; single hindlimb exhibiting abnormal reflex without clasping, 3; both hindlimbs exhibiting abnormal reflex without clasping, 2; both hindlimbs exhibiting abnormal reflex with clasping, 1; both hindlimbs exhibiting abnormal reflex with clasping and forelimb clasping or crossing, 0.

Grip strength was conducted every other week from 4 weeks of age. Mice were held at the base of the tail and pulled across a grip strength apparatus (Ugo Basile, 47200-001) that allowed the mice to grip with all four limbs and peak force was recorded. Mice were tested twice and the higher of the two scores was used for data analysis.

Open field activity monitor analysis was conducted every other week from 4 weeks of age. Mice were placed into an open field activity monitor box (Med Associates, ENV-520). Vertical (vertical counts) and horizontal activity (ambulatory distance) were recorded in 5 min-intervals over a 30 min total duration.

**Preparation of tissue lysates for proteomics and immunoblotting.** The brains of PBS-perfused control and rNLS8 mice were harvested at different timepoints off dox feed, including pre-onset (1 week), onset (2 weeks), early disease (4 weeks), and late disease (6 weeks). Age-matched littermate control mice were either single transgenic hemizygous tetO-hTDP-43-ΔNLS line 4 or hemizygous NEFH-tTA line 8 (Supplementary Data 1). Due to the dox-regulatable expression of hTDP-43ΔNLS, cortex tissue from recovery mice (6 weeks + 2 weeks on dox) was also assessed to determine the proteins involved in functional recovery from hTDP-43ΔNLS-driven disease. For proteomics and

immunoblotting, the brains from 5 mice per group were harvested. The left hemisphere of brains were flash frozen on dry ice and stored at −80 °C. Tissue was suspended in 5 × volume of bicarbonate buffer (100 mM ammonium bicarbonate, 1% sodium deoxycholate, pH 7.6) on ice. Tissues were homogenized with 5 × 1.4 mm zirconium oxide beads (Bertin Instruments) and 3 × 15 s pulses with 30 s pauses at 6000 rpm using a Precellys Evolution (Bertin Instruments) and placed back on ice. Samples were ultracentrifuged at 100,000 × g for 30 min at 4 °C to pellet insoluble protein. The soluble protein fraction supernatant was collected and stored in a new tube. The insoluble protein pellet was washed in bicarbonate buffer and taken through the homogenization and ultracentrifugation steps and the wash supernatant discarded. The urea-soluble fraction was generated by adding urea buffer (7 M urea, 2 M thiourea, 4% CHAPS, 30 mM Tris, pH 8.5) to solubilize the insoluble protein pellet, followed by homogenization and ultracentrifugation.

**Preparation of cell lysates for immunoblotting.** At experiment endpoints, cells were harvested and washed in ice-cold PBS and proteins were extracted in RIPA buffer (50 mM Tris, pH 7.5, 150 mM NaCl, 1% (v/v) NP-40, 0.5% (w/v) sodium deoxycholate, 0.1% (w/v) SDS, 1 mM EDTA, 1 mM PMSF, 1 × protease/phosphatase inhibitor cocktail). Cells were subjected to the same homogenization and ultracentrifugation protocol as described above to wash the insoluble protein pellet and generate a urea-soluble fraction.

RIPA-soluble protein extracts from cortex tissue or cells were quantified using the BCA Protein Quantification Kit (Pierce) and plates were analyzed on a PolarStar Optima plate reader (BMG Lab Technologies). Samples were then analyzed by immunoblotting.

**Tandem mass tag labeling.** Tandem mass tag (TMT) labeling and quantitative proteomics were conducted as previously described[100–102]. Solubilized proteins were reduced by the addition of 0.1 volume of 10 mM dithiothreitol (DTT) for 1 h at room temperature (RT), followed by alkylation with 0.05 volume of 50 mM iodoacetamide (IAA) for 1 h in the dark at RT. To quench the alkylation reaction, 5 mM DTT was added to the sample for 15 min in the dark. Proteins were then digested with trypsin (Promega) 1:100 enzyme-to-protein ratio at 37 °C overnight. The resultant peptides were acidified with 1% trifluoroacetic acid, desalted using styrene divinylbenzene-reverse-phase sulfonate (Empore) stage tips and dried down using a vacuum centrifuge. The dried cortex peptides were solubilized in 100 mM HEPES (pH 8), and the concentration of the final sample was quantified using the BCA Protein Quantification Kit.

For peptide labeling, the TMT10plex Isobaric Label Reagent Set (ThermoFisher Scientific) was used with 100 µg of peptides from each cortex sample as described previously[103]. All steps were conducted at RT. Briefly, 0.8 mg of labeling reagent was added to each sample and the samples were incubated for 1 h with occasional vortexing. Samples were then treated with 8 µL of 5% hydroxylamine for 15 min to quench unbound labeling reagent. To accommodate 45 samples (20 total control samples with 5 mice in 4 groups and 25 total rNLS8 samples with 5 mice in 5 groups), we conducted 5 TMT10plex experiments (Fig. 1a). For each individual TMT10plex experiment 10 uniquely labeled samples were combined and lyophilized in a vacuum centrifuge. Samples were reconstituted in 1% formic acid and desalted using Sep-Pak C18 cartridges (Waters). Peptides were then consolidated into 16 fractions with high pH reverse-phase peptide fractionation as previously described[104]. Dried fractions were resuspended in 1% formic acid and desalted using styrene divinylbenzene - reversed phase sulfonate stage-tips.

**LC-MS/MS analysis.** Samples were analyzed by LC-MS/MS as described previously[103]. Dried peptides were reconstituted in 40 µL of 0.1% formic acid and analyzed using a Q Exactive Orbitrap mass spectrometer (ThermoFisher Scientific) coupled to an EASY-nLC1000

nanoflow high performance liquid chromatography system (Thermo-Fisher Scientific). Reverse-phase chromatographic separation was performed to separate labeled peptides using a Halo 2.7 µm 160 A ES-C18 column (Advanced Material Technology) and a gradient of 1%–30% solvent B (99% acetonitrile and 0.1% formic acid) and Solvent A (97% water, 2% acetonitrile, and 0.1% formic acid). The Q Exactive Orbitrap mass spectrometer was run in data-dependent acquisition mode. The full MS scan spectra (from m/z 350 – 1850) were acquired at a resolution of 70,000 at 400 m/z, an automatic gain control, and a target value of $1 \times 10^6$ ions. The top 10 most abundant ions were selected with a precursor isolation width of 0.7 m/z for higher energy collisional association (HCD). HCD normalized collisional energy was set to 35% and fragmentation ions were detected in the Q Exactive Orbitrap mass spectrometer at a resolution of 70,000. Target ions that had been selected for MS/MS were dynamically excluded for 90 s.

**Database searching.** Raw data files were processed using Proteome Discoverer v2.1, as previously described in[103,105]. Search results were further filtered to retain protein with false discovery rate of <1% and only master proteins assigned via protein grouping algorithm were retained. Relative quantitative ratios to the common reference (label 131, pooled control sample) were extracted, and further analyzed using the TMTPrepPro analysis pipeline[101]. TMTPrepPro scripts are implemented in the R statistical environment and are available as an R package accessed through a graphical user interface provided via a local GenePattern server. The experimental conditions were compared overall, using an analysis of variance (ANOVA) carried out separately for each protein using log-transformed ratios. Pairwise comparison tests were also undertaken for all comparisons of interest using the relevant ratios for each of the conditions of interest (e.g., All controls grouped – TDP Week 1, etc.), and differentially expressed proteins were identified based on Student t tests of log-transformed ratios. The overall fold changes were calculated as geometric means of the respective ratios. Differential expression required the proteins to meet both a ratio fold change (>1.20 up-regulated or <0.83 down-regulated) and a P value cut-off (t test P < 0.05). A combination of fold change and P value cut-off was preferred to limit the false discovery rate over multiple testing corrections due to the known impacts of ratio compression in the context of TMT[106].

**WCNA.** WCNA is a method for gene co-expression network modeling and clustering[107], applied here to group proteins with strongly correlated protein abundance over time[41]. Precise workflow details are available from[41]; The cut-off of the parameter RsquaredCut was set as 0.85 and the soft power chosen for this analysis was 8. A signed network adjacency matrix was constructed by calculating the pairwise correlation between proteins raising to 8. The topology overlap (TOM) distance was then calculated from the network adjacency; the aim of the TOM distance is to identify clusters which are tightly correlated to each other. Hierarchical clustering was performed based on the TOM distance by using average linkage. A set of clusters was obtained by using the dynamic tree cutting method with the parameter value of minClusterSize as 20. Eigenproteins were generated for each cluster as the first principal component by using the singular value decomposition. Hub proteins of each cluster were computed by using the kME value which is the correlation between each protein and its eigenprotein; the higher the kME the more connectivity it has to other proteins in its cluster.

**Proteomics gene ontology analysis.** Protein subsets were deposited into Metascape[108] for inference of enriched biological pathways and protein complexes using default settings. Removal of redundant terms was conducted by filtering for first in class terms.

**Human meta-analysis.** Several relevant and comprehensive proteomics and transcriptomics on human post-mortem cases with

TDP-43 proteinopathies were selected for comparing the results of this study with the disease observations in humans, which provided detailed expression data in a readily usable format[59,60]. Umoh et al.[59], contains proteomics relative abundance and ref. [60], contains transcriptomics differential expression. These selected public data subsets which include log fold changes for various pairwise comparison, significance values, and results of WCNA clustering (where available) were organized into tables, merged, and made available for interactive query as part of a R Shiny web application, publicly available from (https://shiny.rcc.uq.edu.au/TDP-map/). The protein expression data for the current experiment including differentially expressed proteins for all time point comparisons (Supplementary Data 2) and overall differential expression via ANOVA and WCNA (Supplementary Data 5) were similarly processed to retain log fold changes, significance values and cluster membership, and were made available as part of the same Shiny web app. The app enables the easy comparison of the abundance of mouse proteins of interest in the human data, with the mapping of subsets across species using the common gene identifiers. The visualizations provided include UpSet plots for set membership comparisons, pie charts of percentages of differentially expressed proteins in all available comparisons and overall views of abundance in matrix form and volcano plots. In terms of the WCNA clustering, an overview of the cluster membership and data expression across all comparisons is provided for the input proteins, in either the current mouse study or the human proteomics[59] data.

**Human and mouse WCNA cluster comparison.** The clustering of proteins in the current mouse study was compared with the clustering of proteins in the human proteomics dataset of[59], using the available WCNA cluster membership. The comparison aims to determine whether proteins with similar function are clustered similarly in the two species. Data from the two experiments were merged using the gene identifier, and the number and percentages of each cluster combination were tabulated. The association of cluster membership was tested by a Fisher exact test for each combination of clusters from the two datasets, thereby verifying, for example, whether presence in Mouse cluster X is significantly associated with presence in Human cluster Y, for all possible combinations of clusters X and Y. P values adjusted for multiple testing correction (Holm correction) were generated, and values less than 0.001 were regarded as highly indicative of a significant association.

**Cell-type-enrichment analysis of module proteins.** Using the mouse cell-type and brain-region resolved proteome derived by[42], lists of proteins specific to neurons, astrocytes, microglia, and oligodendrocytes were previously generated by[61]. In this study, there were a total of 977 proteins in all modules, which was used as the background. Prior to enrichment analysis, cell-specific lists were filtered for their presence in the total list of module proteins. There were 51 neuron-, 63 astrocyte-, 51 microglia-, and 58 oligodendrocyte-specific proteins identified in the total module protein list. These cell-specific lists were then cross-referenced with the module subsets of proteins to determine whether modules displayed an enrichment for cell type. The enrichment ratio of a cell type within a module was calculated as previously described[109] followed by Fisher's exact test to determine whether the enrichment was statistically significant ($P < 0.05$). The results of the cell-type enrichment analysis were presented using the ComplexHeatmap package (RRID:SCR_017270)[110] in RStudio (Version 1.4.1717; RRID:SCR_000432).

**Tissue processing for immunohistochemistry.** Mice were anesthetized with 2.5 µL/g of sodium pentobarbitol (Virbac). A transcardiac perfusion was then carried out with 25 mL of PBS. The brain was removed, and the right hemisphere was post-fixed in 4% paraformaldehyde (PFA, in PBS; pH 7.2) overnight at 4 °C for downstream histology analysis. Post-fixing, the tissue was washed in excess PBS and the lumbar spinal cord was dissected out of the column. The tissue was then dehydrated in a series of increasing concentrations of ethanol and then embedded in paraffin (Paraplast Plus, Surgipath) using the TES Valida paraffin embedding center (Medite) and Leica TP1020 automatic benchtop tissue processor (Leica Biosystems). The paraffin-embedded brains and lumbar spinal cords were then serially sectioned at 10 µm thickness using a Leica RM2235 manual rotary microtome (Leica Biosystems) and mounted on poly-L-lysine microscope slides (ThermoFisher).

**RNA extraction from tissue for RNA-seq and preparation of cDNA for qPCR.** Flash frozen right rostral cortex tissue was thawed on ice and 1.4 mm zirconium oxide beads were placed into each tube prior to homogenization. Tissue was homogenized in 1 mL QIAzol lysis reagent with $3 \times 15$ s pulses with 30 s pauses at 6000 rpm in a Precellys Evolution tissue homogenizer (Bertin Instruments) and incubated for 5 min at RT to allow foam to dissipate. To separate the aqueous phase, 200 µL of chloroform was added to each tube and shaken vigorously for 15 s, incubated for 2 min at RT, followed by centrifugation at $12,000 \times g$ for 15 min at 4 °C. The aqueous phase containing RNA was transferred to a new tube for extraction and clean-up using the RNeasy Mini Kit (QIAGEN) with an on-column DNase digestion as per the manufacturer's protocol. RNA was eluted from the column in 50 µL of elution buffer. The RNA concentration was quantified using a nano-spectrophotometer (Implen) such that 1 µg of RNA was converted to cDNA using the SensiFast cDNA Synthesis Kit (Meridian Bioscience) as per the manufacturer's protocol.

**Real-time quantitative PCR.** Real-time quantitative PCR was conducted using the SensiFast SYBR No-ROX kit (Meridian Bioscience) according to the manufacturer's instructions. Briefly, SensiFast SYBR No-ROX mix (1×), forward primer (400 nM), reverse primer (400 nM), RNase/DNase-free water and template (up to 20 µL) were mixed per reaction in a 384-well plate. The primers used are listed in the key resources table in the source data file. Plates were centrifuged at $1000 \times g$ for 1 min at RT prior to loading on a LightCycler 480 (Roche) using 3-step cycling. All reactions were conducted as triplicates of five biological replicates ($n = 5$). Fold changes in mRNA transcripts were calculated using the $2^{-\Delta\Delta CT}$ method relative to the mean of control mice and normalized to the *Gapdh* gene.

**Transfection.** HEK293 cells were transfected using Lipofectamine 2000 (Invitrogen) and plasmid DNA purified with the Plasmid Plus Maxi Kit (Qiagen) according to the manufacturer's instructions. HEK293 cells were seeded onto coverslips in 24-well plates at a density of 100,000 cells per mL and incubated for 24 h. For transfections, 0.75 µg DNA and 2.25 µL Lipofectamine 2000 were mixed in a total of 150 µL Opti-MEM (Gibco) and incubated for 5 min before dropwise addition to cells. Co-transfections involving two plasmids were conducted at a 1:1 ratio (375 ng DNA 1: 375 ng DNA 2). Cells were incubated for 48 h to allow protein expression from the plasmid DNA before analysis by immunoblot or immunocytochemistry.

For CRISPR knockout experiments, HEK293 cells stably expressing Cas9 endonuclease (HEK293:Cas9) were co-transfected as described above with plasmids for the expression of TDP-43 (DNA 1) and an equimolar mix of 3 plasmids coding for unique *DNAJB5* sgRNA (DNA 2; see key resources table in the source data file). Cells were harvested 72 h post-transfection.

**Lentiviral preparation and transduction of primary cortical neurons.** HEK293 cells were transfected to produce lentiviral particles with TransIT-Lenti transfection reagent (Mirus) according to the manufacturer's instructions. Briefly, in a 6-well plate, 1 µg transfer DNA was mixed with 1 µg total packaging DNA mix (1:1:1, pMDLg/pRRE:

pRSV-Rev: pMD2.G) and 6 µL TransIT-Lenti reagent in 200 µL Opti-MEM and incubated at RT for 10 min. The mix was applied dropwise to a 90% confluent cell monolayer and incubated for 48 h prior to harvest. Medium (2 mL) containing lentivirus was collected and centrifuged at 1000 × g for 10 min at 4 °C to remove cell debris. Lentivirus was concentrated by ultracentrifugation on an Optima MAX-XP (Beckman) at 75,000 × g for 2 h at 4 °C, and the lentivirus pellet was resuspended in 200 µL neurobasal medium with rocking overnight at 4 °C, then aliquoted and snap frozen with liquid nitrogen.

After 7 days in vitro, primary cortical neurons were treated with lentivirus to express pLenti-TDP-43$^{\Delta NLS/2KQ}$-mGFP in combination with pLenti-hSyn1-DNAJB5-FLAG or pLenti-hSyn1-EGFPi-FLAG (negative control plasmid), then fixed and immunolabeled after 7 days of expression.

**Immunoprecipitation.** HEK293 cells were harvested after 48 h of transfection in Beckman Coulter microfuge tubes by centrifugation at 300 × g for 5 min in PBS. Cell pellets lysed in RIPA buffer. The cells were homogenized using a Precellys Evolution homogeniser using the following settings: 6000 rpm, 3 × 15 s pulses with 30 s pauses. After homogenization, the cells were centrifuged in Beckman Coulter Optima MAX-XP at 100,000 × g for 30 min at 4 °C. The detergent soluble protein fraction was quantified using the Pierce BCA Protein Assay Kit, as per manufacturer's instructions. The immunoprecipitation of tagged proteins was performed using 50 ml Dynabead Protein A magnetic beads. The beads were washed 1 x with PBS buffer before incubating in GFP or FLAG antibodies. The corresponding antibody dilutions were made in 400 ml PBS + 0.05% Tween-20 buffer and incubated with the beads for 20 mins at RT on gentle rotation. The flow-through after antibody binding were discarded and the beads were washed 3 x in PBS + 0.05% Tween-20. The washed beads were incubated with 1 mg of 1 ml protein lysate at 4 °C overnight on gentle rotation. The flow through after lysate binding was discarded and the beads were washed 3 x with 500 µL RIPA (300 mM NaCl) buffer for 5 min each. The beads were then washed once with 500 µL RIPA (150 mM NaCl) buffer for 5 min and transferred to a fresh microcentrifuge tube. The wash buffer was then discarded, and the beads were added with 30 µL of 2 × Laemmli buffer with 5% (v/v) β-mercaptoethanol. The samples were then incubated at 70 °C for 10 min to elute immunoprecipitated proteins. The eluted proteins were run along with their input protein lysate on 8% SDS-PAGE gel and immunoblotted (detailed in the immunoblot section).

**Immunoblotting.** Samples were all made up to 1 mg/mL in RIPA buffer in reducing Laemmli buffer supplemented with 5% (v/v) β -mercaptoethanol. Samples (10 µg per well) were loaded on 12% SDS-PAGE gels and run at 150 V for 60 min, and transferred to a nitrocellulose membrane (Li-Cor) at 100 V for 90 min. Total protein was quantified with Revert 700 total protein stain (Li-Cor) according to the manufacturer's instructions and then blocked in 5% (w/v) BSA Tris-buffered saline (0.5 M Tris-base, 1.5 M NaCl) with 0.05% (v/v) Tween-20 (TBS-T). Blots were immunolabeled overnight at 4 °C with primary antibodies (see key resources table in the Source Data file for antibodies used, catalog numbers, and concentrations). Blots were washed 4 x in TBS-T and incubated in secondary antibodies for 1 h at room temperature. Blots were washed again 4 x in TBS-T and then imaged using the Odyssey Clx imaging system (Li-Cor). Immunoblot images were analyzed using Image Studio software (Li-Cor). All uncropped immunoblot scans are included in the Source Data file and at the end of the Supplementary Information document.

**Immunocytochemistry.** At experiment endpoints, cells on coverslips were fixed with 4% (w/v) PFA in PBS for 15 min at RT. Cells were permeabilized and blocked by incubation with 5% BSA (w/v) in PBS with 0.1% Triton X-100 for 1 h at RT and then immunolabeled with anti-FLAG

(1:1000, F1804, Sigma) overnight at room temperature. The coverslips were washed 3 x in PBS and labeled with donkey anti-mouse AlexaFluor 647 (1:1000, A31571, Thermo Fisher), for 1 h at RT. The coverslips were then stained with DAPI (1 µg/mL, Thermo Fisher) for 10 min at RT and then washed 3 × in PBS. Coverslips were mounted onto microscope slides using Prolong Diamond Antifade mountant (P36961, Thermo Fisher) and allowed to cure for 24 h at RT before microscopy.

**Immunofluorescence of cortex tissue sections.** PFA fixed, paraffin-embedded sections from cortex tissue were de-waxed by incubation in xylene for 15 min, followed by a second 15 min xylene incubation, 100% ethanol for 5 min, a second incubation in 100% ethanol for 5 min, 70% ethanol for 5 min, and rehydration in water for 5 min. The rehydrated slides were placed in antigen retrieval buffer (10 mM sodium citrate, 0.05% Tween-20, pH 6.0) for heat-induced antigen retrieval at 95 °C for 10 min in the Decloaking Chamber NxGen (Biocare Medical). Sections were blocked (5% w/v BSA, 0.25% v/v Triton X-100, in 1 x PBS) for 20 min at RT, and incubated with primary antibodies (1% BSA, 0.25% Triton X-100, 1 x PBS) overnight at RT in a humidified chamber. They were then washed in PBS and blocked in 5% BSA PBS for 5 min prior to incubation with secondary antibodies in 1% BSA PBS for 1.5 h at RT. Slides were washed in PBS, rinsed in water, counterstained with DAPI in PBS (1 µg/mL, Thermo Fisher) for 5 min, and rinsed a final time with PBS before mounting with fluorescence mounting medium (Dako). The slides were then imaged using widefield or confocal microscopy.

**Widefield fluorescence microscopy.** Widefield fluorescence images were captured using a Zeiss Axio Imager with a 20 x air objective (0.5 NA, 2.0 mm WD, 0.323 µm/pixel) with Colibri LEDs for excitation of DAPI (365 nm), AlexaFluor488/mGFP (470 nm), AlexaFluor594 (590 nm), AlexaFluor647 (625 nm). Fluorescence emissions were collected using the following filter sets; DAPI (BP 420–470 nm), AlexaFluor488/mGFP (BP 505–555 nm), AlexaFluor594 (BP 570–640 nm) and AlexaFluor647 (BP 665–715 nm). Images were acquired using Zen software and all acquisition parameters were set using single color and secondary antibody only controls and kept the same for all samples in an experimental set. Original raw images were exported for downstream analyses and Zen software was used to derive display settings that were applied consistently across all images within an experimental set.

**Confocal microscopy.** Confocal images were captured using either the Zeiss Laser Scanning Microscope (LSM) 510 META or Zeiss LSM 710. Both were equipped with a 63x oil objective (1.4 NA, 190 µm WD, 0.19 µm/pixel), blue laser diode for DAPI (405 nm), argon laser for AlexaFluor488 (488 nm), 561 nm laser for AlexaFluor555, and HeNe 633 nm laser for AlexaFluor647 excitation. Images were acquired using Zen software and all acquisition parameters were applied consistently for all samples in an experimental set.

**Image analysis.** All image analyses was conducted using CellProfiler Image Analysis Software (version 4.2.1, RRID:SCR_007358)[111] using custom pipelines that are available on request.

**Multiplexed fluorescence immunohistochemistry and image acquisition of human post-mortem tissue.** Fluorescence immuno-histochemistry of 10 µm thick sections of formalin-fixed paraffin-embedded motor cortex from post-mortem human donors (Table 2) to the Neurological Foundation Human Brain Bank at the Centre for Brain Research, University of Auckland was performed as described previously[21,112,113]. Briefly, following dewaxing and rehydration in ethanol, antigen retrieval was performed for 2 h in citrate buffer (10 mM citrate, 0.05% Tween 20, pH 6.0) and tissue permeabilized with phosphate-buffered saline with 0.1% Triton X-100 (PBS-T) for 15 min at 4 °C. Autofluorescence was eliminated using TrueBlack (Biotium, 1:20

in 70% ethanol) for 30 s, and sections blocked in 10% normal goat serum (NGS) in PBS-T for 1 h, both at RT. Sections were incubated with primary antibodies overnight at 4 °C and secondary antibodies for 3 h at RT, both in immunobuffer (PBS-T + 1% NGS, antibodies in key resources table in the source data file). Nuclei were counterstained using Hoechst 33258 (Sigma) at 1:2000.

Wide-field images were acquired using a Zeiss Z2 Axioimager with a MetaSystems VSlide slide scanning microscope (20x dry magnification lens, 0.9 NA) with a Colibri 7 solid-state fluorescent light source. Filters were as described and validated previously[114]. Exposure times were consistent within each channel between cases.

Figures were compiled using FIJI software (v1.53c, National Institutes of Health) and Adobe Illustrator (Adobe Systems Incorporated, v24.3). Extracted single-channel images were imported into FIJI, merged, and pseudocoloured. Image intensities for Hoechst and phosphoTDP-43 were consistent across cases but for DNAJB5 were adjusted in FIJI to best represent each case. All image intensities were set to best view the marker of interest, reduce background autofluorescence, and generate representative images.

### Statistics and reproducibility

For animal experiments for biochemical analysis, all experiments were conducted with 5 mice per group. Cell experiments were conducted over 3 independent replicates. Individual data points for each biological replicate (mouse) or independent replicate (experiment replicate) are shown overlaid with floating bars (min to max) with a line at the mean. Statistically significant differences between the means were determined using a two-tailed paired *t* test (where indicated in the figure legend) for comparisons of control versus disease. For comparisons across more than two groups, a one-way ANOVA was used followed by a Tukey's post hoc test for multiple comparisons. All statistical analyses were conducted using GraphPad Prism (RRID:SCR_002798, version 9.4.1, GraphPad Software). The Investigators were blinded to allocation of mice during experiments and behavioral outcome assessment.

**Group sizes**. Group sizes of 5 control and 5 bigenic rNLS8 mice per timepoint were used for quantitative proteomics, immunoblotting, qPCR, and immunohistochemistry. Cortex tissue was collected from control and rNLS8 mice at pre-onset (1 wk off dox), onset (2 wk), late-disease (6 wk) and recovery (6 wk off dox followed by 2 wk on dox to stop hTDP-43$^{\Delta NLS}$ expression). If mice were lost from a study for reasons unrelated to the development of disease, the data from these mice were removed from all analyses.

### Additional resources

Selected public data subsets[59,60] and the proteomics dataset produced in this research, including log fold changes for various pairwise comparison, significance values and results of WCNA clustering where made available for interactive query as part of a R Shiny web application, publicly available from [https://shiny.rcc.uq.edu.au/TDP-map/].

### Reporting summary

Further information on research design is available in the Nature Portfolio Reporting Summary linked to this article.

## Data availability

The mass spectrometry proteomics data have been deposited to the ProteomeXchange Consortium via the PRIDE[115] partner repository with the dataset identifier PXD042382. Previously published human proteomics[59] and transcriptomics[60] datasets used in this work for comparative analyses are available from the respective references. Source data and Supplementary Data are provided with this paper and any additional information required to re-analyze the data reported in this paper is available from the lead contact upon request. Source data are provided with this paper.

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

## Acknowledgements

The authors gratefully acknowledge the Queensland Brain Institute's Advanced Microscopy Facility for their support and assistance in this work. Imaging was performed using Zeiss LSM 510 and LSM 710 microscopes, generously supported by the Australian Government through the ARC LIEF grant LE130100078. We wish to acknowledge The University of Queensland's Research Computing Centre (RCC) for support of this research. We thank Rowan Tweedale and members of the Neurodegeneration Pathobiology Laboratory for their critical evaluation of the manuscript. We acknowledge insightful discussions relating to this work with the late Professor Justin Yerbury. All schematics were created with BioRender.com. R.S.G. was supported by a FightMND Early Career Research Fellowship. This work was supported by the National Health and Medical Research Council (RD Wright Career Development Fellowship 1140386), the FightMND Bill Guest Mid-Career Research Fellowship, FightMND IMPACT grant, the Ross Maclean Fellowship, the Brazil Family Program for Neurology, and Motor Neuron Disease Research Australia (IG2062) to A.K.W.. S.S.K. was supported by a Thornton Foundation PhD Scholarship and a MND Research Australia PhD Scholarship top-up scholarship. A.T.B. was supported a Race Against Dementia–Dementia Australia Research Foundation Fellowship. S.T.N. was supported by a FightMND Mid-Career Research Fellowship.

## Author contributions

R. San Gil and AKW conceived the research; R. San Gil, DP, HBW, PM, JW, AL, M. Morsch, R.S. Chung, LL, M. Mirzaei and AKW contributed to design of the project. R. San Gil, JV, HBW, PM, LMSM, WL, YKC, ATB, SS, SN, BAB, ALW, SSK, JDL, STN, ES, M. Mirzaei and AKW performed experiments; RLMF and MAC collected and processed the human tissue. R. San Gil, DP and JW analyzed the data. DP constructed the Shiny webapp. R. San Gil prepared figures. R. San Gil and AKW interpreted the results and wrote the manuscript. All authors read and approved the final manuscript.

## Competing interests

The authors declare no competing interests.

## Additional information

[1]Neurodegeneration Pathobiology Laboratory, Clem Jones Centre for Ageing Dementia Research, Queensland Brain Institute, The University of Queensland, Brisbane, QLD, Australia. [2]Insight Stats, Croydon Park, NSW, Australia. [3]Macquarie Medical School, Faculty of Medicine, Health and Human Sciences, Macquarie University, North Ryde Sydney, NSW, Australia. [4]Motor Neuron Disease Research Centre, Macquarie Medical School, Macquarie University, Sydney, NSW, Australia. [5]School of Biological Sciences, University of Auckland, Auckland, New Zealand. [6]Centre for Brain Research, University of Auckland, Auckland, New Zealand. [7]Department of Anatomy and Medical Imaging, University of Auckland, Auckland, New Zealand. [8]School of Biomedical Sciences, Faculty of Medicine, The University of Queensland, St Lucia, Brisbane, QLD, Australia. [9]Australian Institute for Bioengineering and Nanotechnology, The University of Queensland, Brisbane, QLD, Australia. [10]Vector and Genome Engineering Facility, Children's Medical Research Institute, Westmead, NSW, Australia. [11]Laboratory of Molecular Oncology and Innovative Therapies, Military Institute of Medicine – National Research Institute, Warsaw, Poland. [12]Translational Vectorology Research Unit, Children's Medical Research Institute, Faculty of Medicine and Health, The University of Sydney, Westmead, NSW, Australia. ✉e-mail: adam.walker@uq.edu.au

