## [Peer Review File · Nature Communications]

A transient protein folding response targets aggregation in the early phase of TDP-43-mediated diseaseEditorial Note: This manuscript has been previously reviewed at another journal that is not operating a transparent peer review scheme. This document only contains reviewer comments and rebuttal letters for versions considered at *Nature Communications*.

REVIEWER COMMENTS

Reviewer #1 (Remarks to the Author):

The authors have made a good faith effort to address prior concerns and have tried to address various points with data. Overall, this paper will make a valuable addition to the literature. I recommend publication.

Reviewer #2 (Remarks to the Author):

The authors have addressed the majority of my concerns, through new experiments, modifications to the text, or both. The revised manuscript is significantly improved and a good fit for Nature Communications.

Reviewer #3 (Remarks to the Author):

Rebecca San Gil and colleagues have made a significant effort to address the numerous comments/suggestions raised by the three reviewers including some key points that were essential to establish the relevance of DNAJB5 in TDP-43 proteinopathy. In particular the authors now provide new evidence that 1) overexpression of a TDP-43 variant (TDP-43deltaNLS/2KQ-mGFP, cytoplasmic acetylation-mimetic mutant) in Hek cells lacking DNAJB5 leads to increased insoluble P-TDP-43 and 2) that both proteins interact (revealed by an immunoprecipitation assay in cells overexpressing both transgenes). These are additions that support the potential involvement of DNAJB5 in TDP-43 aggregation.

That said, some confusion remains as these key experiments were performed uniquely in the context of overexpression of a TDP-43 variant that is not the one that was most studied in the paper (TDP-43deltaNLS), and whose relevance for human pathogenesis is not established. Are these findings also recapitulated in the context of TDP-43DeltaNLS or more importantly, in more disease-relevant mutants when expressed in Hek cells and neurons? Given that most of the work provided was performed in the context of TDP-43 deltaNLS, it would be important to also demonstrate that there is increased insolubility/levels/aggregation in TDP-43deltaNLS expressing cells, which is more widely used, but also in cells expressing ALS-linked mutants.

The authors did try to address that in mice, but the outcome of the findings when TDP-43deltaNLS or TDP-43Q331K were analyzed does not fully support the conclusion that would be widely applicable to all TDP-43 forms. Indeed, no exacerbation of TDP-43 pathology nor behavioral deficits were observed in

Dnajb5 KO mice which overexpress TDP-43deltaNLS using AAVs (after 3 months) in the new assays provided (new figure Sup Fig 20 and 21). It would be important to re-evaluate these key parameters at later time-points (>6 months of age).

Therefore, to strengthen the findings the authors are encouraged to validate the relevance of DNAJB5 in human autopsied material as previously requested.

It remains unclear whether the changes observed are due to gain of toxicity as proposed here and/or loss of function of nuclear TDP-43. The authors propose that the reason why no changes are observed in the context of TDP-43Q331K compared to DeltaNLS is due to the lack of cytoplasmic mislocalization/aggregation of TDP-43 in this model. This is indeed a possibility. Yet it could also be due to the absence of loss of TDP-43 function as pointed out by the reviewers.

Minor additions:

- What is the % of variance driven by PC1, PC2, PC3?
- There is a typo in the word TDP-43DeltaNLS-myc in the legend of SUP Fig 20.
- Sup Fig 14 is a key figure and yet it is very difficult to read it as it is too small. Perhaps the figure could occupy the full page.
- The title in the legend of Sup Fig21 should be modified. It states: “ Dnajb5 KO does not exacerbate.....TDP-43delta NLS, mouse model of ALS.” This is not a mouse model of ALS so this sentence should be rephrased to avoid this overstatement about the model.

REVIEWER COMMENTS

Reviewer comments in *blue italics*, and our author responses in **red non-italicised**, referring to page/line numbers with amended text highlighted in the updated, red-lined version):

Reviewer #1 (Remarks to the Author):

The authors have made a good faith effort to address prior concerns and have tried to address various points with data. Overall, this paper will make a valuable addition to the literature. I recommend publication.

Reviewer #2 (Remarks to the Author):

The authors have addressed the majority of my concerns, through new experiments, modifications to the text, or both. The revised manuscript is significantly improved and a good fit for Nature Communications.

Reviewer #3 (Remarks to the Author):

Rebecca San Gil and colleagues have made a significant effort to address the numerous comments/suggestions raised by the three reviewers including some key points that were essential to establish the relevance of DNAJB5 in TDP-43 proteinopathy. In particular the authors now provide new evidence that 1) overexpression of a TDP-43 variant (TDP-43deltaNLS/2KQ-mGFP, cytoplasmic acetylation-mimetic mutant) in Hek cells lacking DNAJB5 leads to increased insoluble P-TDP-43 and 2) that both proteins interact (revealed by an immunoprecipitation assay in cells overexpressing both transgenes). These are additions that support the potential involvement of DNAJB5 in TDP-43 aggregation.

We thank the reviewer for their comments and have both included new text in the manuscript and conducted additional experiments to support the existing data to address the points below.

That said, some confusion remains as these key experiments were performed uniquely in the context of overexpression of a TDP-43 variant that is not the one that was most studied in the paper (TDP-43deltaNLS), and whose relevance for human pathogenesis is not established. Are these findings also recapitulated in the context of TDP-43DeltaNLS or more importantly, in more disease-relevant mutants when expressed in Hek cells and neurons? Given that most of the work provided was performed in the context of TDP-43 deltaNLS, it would be important to also demonstrate that there is increased insolubility/levels/aggregation in TDP-43deltaNLS expressing cells, which is more widely used, but also in cells expressing ALS-linked mutants.

The focus of this work was to interrogate the molecular mediators of TDP-43 pathology in models of sporadic disease, which represents the form of the vast majority of ALS with TDP-43 pathology and also FTLD-TDP. Therefore, the major models used were

TDP-43 with a defective nuclear localisation signal to recapitulate cytoplasmic accumulation and formation of puncta, with a lesser focus on the rare ALS-linked *TARDBP* mutants. We believe that our focus on the cytoplasmic form of TDP-43 and not on the rare ALS-linked mutants of TDP-43 is valid in the context of understanding sporadic disease. We have now also provided further justification for the use of TDP-43^{ΔNLS/2KQ}, which mimics the acetylation post-translational modifications previously identified in the spinal cord of sporadic ALS patients, to provide clear links to human ALS pathogenesis.

- New text, page 20, lines 3-4: "...cytoplasmic mutant (TDP-43^{ΔNLS/2KQ}) that mimics acetylation previously identified in spinal cord tissue of ALS cases⁵⁰..."

Regarding the cell models of cytoplasmic TDP-43, we have conducted new experiments that have similarly shown increased insoluble TDP-43 levels in the context of *DNAJB5* knockout with two different cytoplasmic variants of TDP-43 (new data showing results with TDP-43^{ΔNLS} now added, in addition to the previously included TDP-43^{ΔNLS/2KQ}).

- See new data in Supplementary Figure 17.
- New text, page 20, lines 18-21: "Furthermore, CRISPR knockout of *DNAJB5* in HEK293 Cas9 stable cells over-expressing TDP-43^{ΔNLS}-mGFP or TDP-43^{ΔNLS/2KQ}-mGFP resulted in a significant increase in insoluble TDP-43 and phosphoTDP-43 (Supplementary Figure 17 and 18), suggesting that endogenous DNJAB5 is important in mediating the pathological states of cytoplasmic TDP-43."

The authors did try to address that in mice, but the outcome of the findings when TDP-43deltaNLS or TDP-43Q331K were analyzed does not fully support the conclusion that would be widely applicable to all TDP-43 forms. Indeed, no exacerbation of TDP-43 pathology nor behavioral deficits were observed in Dnajib5 KO mice which overexpress TDP-43deltaNLS using AAVs (after 3 months) in the new assays provided (new figure Sup Fig 20 and 21). It would be important to re-evaluate these key parameters at later time-points (>6 months of age).

We agree, that the AAV9 model of cytoplasmic TDP-43 pathology at 3 months likely represents an early diseases stage and have recommended that investigations at later disease stage is warranted in future work, but we believe this is beyond the scope of the present study.

- Page 22, lines 31 onwards: "At 3 months there was no significant difference in cortex soluble or insoluble levels of TDP-43 between WT, heterozygous, or homozygous *Dnajib5* knockout mice injected with TDP-43^{ΔNLS}-myc (Supplementary Figure 22), however investigations at later disease stages such as 6 months are warranted."

We do wish to emphasise however, that our data demonstrate that *Dnajib5* KO mice did develop exacerbated behavioural deficits with overexpression of TDP-43^{ΔNLS} (compared to *Dnajib5* WT mice), including faster decline in hindlimb splay phenotype

(Figure 5) and impairment in grip strength (Supplementary Figure 21) by the 3 month timepoint examined, although we acknowledge that other features of later disease such as weight loss and rotarod decline were not affected by Dnajb5 KO at this timepoint.

Therefore, to strengthen the findings the authors are encouraged to validate the relevance of DNAJB5 in human autopsied material as previously requested.

We believe a major strength of the revised manuscript is the addition of new data from human post-mortem motor cortex tissue analysis of control and ALS cases with phosphorylated TDP-43 pathology. At the request of the reviewer, we have performed immunofluorescence analysis of DNAJB5 levels and subcellular localisation in neurons of the motor cortex in $n = 3$ control and $n = 3$ ALS cases, with case demographics and clinical features added to the manuscript. As expected, based on the transient early increase in DNAJB5 protein levels in the rNLS8 TDP-43 mice pre-onset and at onset, we observed no difference in the levels of diffuse DNAJB5 staining between control and disease in human tissues. However, in support of the conclusion that DNAJB5 plays a role in disease, DNAJB5 partially co-localised with phosphoTDP-43 inclusions.

- New Table 2. Demographics and clinical features of human brain donors.
- New figure 6. Expression of DNAJB5 and phosphoTDP-43 in the post-mortem motor cortex of human ALS and ALS-FTD cases.
- New text, page 26, lines 1-10: *“DNAJB5 distribution is altered in neurons with TDP-43 pathology in human autopsy tissues: We next sought to determine the levels and cellular localization of DNAJB5 in the human motor cortex of non-neurological disease controls and ALS and ALS + FTD cases (Table 2). In post-mortem human brain tissue, DNAJB5 was expressed at a low level in neurons of the motor cortex with no apparent difference in levels between controls and disease cases (Figure 6). However, in cases with ALS or ALS + FTD, DNAJB5 appeared enriched perinuclearly in neurons containing perinuclear phosphoTDP-43 aggregates (Figure 6), suggesting sequestration of DNAJB5 with TDP-43 inclusions.”*
- New text, Abstract page 2, line 7: *“...DNAJB5, which similarly co-localized with TDP-43 pathology in human motor cortex.”*
- New text, Introduction page 3, lines 25-26: *“...DNAJB5... and co-localized with TDP-43 inclusions in human motor cortex.”*
- New methods, page 53. Multiplexed fluorescence immunohistochemistry and image acquisition of human post-mortem tissue

It remains unclear whether the changes observed are due to gain of toxicity as proposed here and/or loss of function of nuclear TDP-43. The authors propose that the reason why no changes are observed in the context of TDP-43Q331K compared to DeltaNLS is due to the lack of cytoplasmic mislocalization/aggregation of TDP-43 in this model. This is indeed a possibility. Yet it could also be due to the absence of loss of TDP-43 function as pointed out by the reviewers.

Regarding the rNLS8 and TDP-43^{Q331K} comparison, we have amended the text to address both loss-of-function and gain-of-toxicity associated with the pathobiology of ALS that is captured by each of these models.

- New text, pages 16-17, lines 27 onwards: “The differences in protein folding responses between the rNLS8 and TDP-43Q331K mice is not surprising, given that they may model different aspects of the pathobiology of ALS. The TDP-43Q331K mouse shows loss of normal TDP-43 function (aberrant splicing) but not cytoplasmic TDP-43 inclusions and developed late-onset motor deficits that do not progress to a typical ALS-like disease end-stage^{46,48}. In contrast, the rNLS8 TDP-43 mouse shows a decrease in endogenous TDP-43, suggesting potential loss of normal nuclear function, in addition to cytoplasmic TDP-43 inclusions, neurodegeneration, neuroinflammation, and shortened lifespan¹⁸, however, whether aberrant splicing of TDP-43 target transcripts occurs in rNLS8 mice remains to be established.”

Minor additions:

- *What is the % of variance driven by PC1, PC2, PC3?*

- *There is a typo in the word TDP-43DeltaNLS-myc in the legend of SUP Fig 20.*

- *Sup Fig 14 is a key figure and yet it is very difficult to read it as it is too small.*

Perhaps the figure could occupy the full page.

- *The title in the legend of Sup Fig21 should be modified. It states: “ Dnajb5 KO does not exacerbate.....TDP-43delta NLS, mouse model of ALS.” This is not a mouse model of ALS so this sentence should be rephrased to avoid this overstatement about the model.*

We have addressed each of these points throughout the text and in all figures and supplementary figures.

REVIEWERS' COMMENTS

Reviewer #3 (Remarks to the Author):

The authors have addressed my main concerns including addition of new experiments which were key in my view. The revised manuscript is much improved and is of interest to the field, making it competitive for Nature Communications.

One minor comment:

- page 16 line 29: it is stated that "The TDP-43Q331K mouse shows loss of normal TDP-43 function(aberrant splicing) but not cytoplasmic TDP-43 inclusions...".

This sentence should be revised as the original study cited provided evidence that TDP-43Q331K does induce indeed aberrant splicing but through both a gain and loss of function(and not loss of function only, as stated).

REVIEWER COMMENTS

Statement with our response (reviewer comments in *blue italics*, and our author responses in **red non-italicised**, referring to page/line numbers with amended text highlighted in the updated, red-lined version):

Reviewer #3 (Remarks to the Author):

The authors have addressed my main concerns including addition of new experiments which were key in my view. The revised manuscript is much improved and is of interest to the field, making it competitive for Nature Communications.

One minor comment:

- page 16 line 29: it is stated that "The TDP-43Q331K mouse shows loss of normal TDP-43 function(aberrant splicing) but not cytoplasmic TDP-43 inclusions...". This sentence should be revised as the original study cited provided evidence that TDP-43Q331K does induce indeed aberrant splicing but through both a gain and loss of function(and not loss of function only, as stated).

We thank the reviewer for their constructive peer review, which has enabled us to produce an improved manuscript.

We have now made this minor correction to the text to better reflect the original cited research:

- **Page 12, lines 21-23: The TDP-43^{Q331K} mouse shows **gain and** loss of normal TDP-43 function (aberrant splicing) but not cytoplasmic TDP-43 inclusions and developed late-onset motor deficits that do not progress to a typical ALS-like disease end-stage^{46, 48}.**